



**Process studies at the air-sea interface after atmospheric deposition in the Mediterranean**
**Sea: objectives and strategy of the PEACETIME oceanographic campaign (May-June**
**2017)**
Cécile Guieu[1], Fabrizio D'Ortenzio[1], François Dulac[2], Vincent Taillandier[1], Andrea Doglioli[3],
Anne Petrenko[3], Stéphanie Barrillon[3], Marc Mallet[4], Pierre Nabat[4], Karine Desboeufs[5]
[1] CNRS, Sorbonne Université, Laboratoire d'Océanographie de Villefranche, UMR7093,
Villefranche-sur-Mer, France
[2] Laboratoire des Sciences du Climat et de l'Environnement (LSCE), UMR 8212, CEA-CNRS-
UVSQ, IPSL, Univ. Paris-Saclay, CEA Saclay, Gif-sur-Yvette
[3] Aix-Marseille Université, CNRS, Université de Toulon, IRD, Mediterranean Institute of
Oceanography, UMR 7294, Marseille, France
4 Centre National de Recherches Météorologiques, Météo-France/CNRM/GMGEC/MOSCA,
Toulouse, France
[5] LISA (Laboratoire Interuniversitaire des Systèmes Atmosphériques), UMR CNRS 7583,
Université de Paris, Université Paris Est, IPSL, Créteil, France
**Abstract**
In spring, the Mediterranean Sea, a well-stratified low nutrient low chlorophyll region, receives
atmospheric deposition both desert dust from the Sahara and airborne particles from
anthropogenic sources. Such deposition translates into a supply of new nutrients and trace
metals for the surface waters that likely impact biogeochemical cycles. However, the
quantification of the impacts and the processes involved are still far from being assessed in situ.
In this paper, we provide a state of the art regarding dust deposition and its impact on the
Mediterranean Sea biogeochemistry and we describe in this context the objectives and strategy





of the PEACETIME project and cruise, entirely dedicated to filling this knowledge gap. Our
strategy to go a step forward than in previous approaches in understanding these impacts by
catching a real deposition event at sea is detailed. The PEACETIME oceanographic campaign
took place in May-June 2017 and we describe how we were able to successfully adapt the
planned transect in order to sample a Saharan dust deposition event, thanks to a dedicated
strategy, so-called 'Fast Action'. That was successful, providing, for the first time in our
knowledge, a coupled atmospheric and oceanographic sampling before, during and after an
atmospheric deposition event. Atmospheric and marine in situ observations and process studies
have been conducted in contrasted area and we summarize the work performed at sea, the type
of data acquired and their valorization in the papers published in the special issue.
1.  **Introduction**
Understanding the exchange of energy, gases and particles at the ocean–atmosphere interface
is critical for the development of robust predictions of future climate change and its
consequences on marine ecosystems and the services they provide to society. Our
understanding of such exchanges has advanced rapidly over the past decade but we remain
unable to adequately parameterize fundamental controlling processes as identified in the new
research strategies of the international Surface Ocean–Lower Atmosphere Study group (Law et
al., 2013 and SOLAS 2015-2025: Science Plan and Organisation). A critical bottleneck is the
parameterization and representation of the key processes brought into play by atmospheric
deposition in Low Nutrient Low Chlorophyll (LNLC) regions. A perfect example of a LNLC
region, and of the role of the atmospheric deposition, is the Mediterranean Sea where the
ecosystem functioning may be modulated by pulsed atmospheric inputs in particular the
deposition of Saharan dust (Guieu et al., 2014a) and nutrients of anthropogenic origin (Richon
et al., 2018a, 2018b).





Indeed, the Mediterranean quasi-enclosed basin continuously receives anthropogenic aerosols
originating from industrial and domestic activities from all around the basin and other parts of
Europe, both in the western (Bergametti et al., 1989; Desboeufs et al., 2018) and eastern
(Tsapakis et al., 2006; Moon et al., 2016) basin. In addition to this continuous 'background'
inputs, the surface of the Mediterranean Sea episodically receives biomass burning particles
(Guieu et al., 2005) and Saharan dust (e.g. Loÿe-Pilot et al., 1986, Vincent et al., 2016). Some
deposition events are qualified as 'extreme events', as dust inputs as high as 22 g m$^{-2}$ (event in
Nov. 2001 recorded at Ostriconi-Corsica Island, Guieu et al., 2010; event in Feb. 2002 recorded
at Cap Ferrat, Bonnet and Guieu, 2006) can occur on very short time scales (hours to days)
representing the main annual dust flux. Associated atmospheric deposition of major macro-
nutrient (N, P) (Kouvarakis et al., 2001; Markaki et al., 2003, 2010; Guieu et al., 2010), of iron
(Bonnet and Guieu, 2006) and of trace metals (Theodesi et al., 2010; Guieu et al., 2010;
Desboeufs et al., 2018) represents significant inputs likely supporting the primary production
in surface waters especially during the stratification period (Richon et al., 2018a, 2018b).
Among the atmospheric deposited nutrients, anthropogenic reactive nitrogen is critical on the
fluxes of inorganic and organic N (Markaki et al., 2010, Violaki et al., 2010). Soil dust
deposition plays an important role on the fluxes of P and trace metals due to the intense but
sporadic inputs (Bergametti et al., 1992; Guieu et al., 2010; Morales-Baquero and Perez-
Martinez, 2016), even if the contribution of anthropogenic aerosol deposition is significant
(between less of 10 % (Fe) and 90% (Zn)) (Guieu et al., 2010, Desboeufs et al., 2018). The
atmospheric deposition of mineral dust is correlated with dissolved trace metals enrichment of
the sea-surface microlayer (Cd, Co, Cu, Fe) (Tovar Sanchez et al., 2014). However, it has been
shown that dust deposition can result either in a net release or in scavenging of dissolved
inorganic phosphorus and nitrate (Louis et al., 2015) and trace elements in seawater (Wagener





et al., 2010; Wuttig et al., 2013; Bressac & Guieu 2013), depending on the quantity and quality
of in situ dissolved organic matter at the time of the deposition.
Recent studies in pelagic large mesocosms also allowing quantifying the export below, have
shown that wet Saharan dust analog deposition, by providing P and N for marine biosphere,
strongly stimulates primary production and phytoplanktonic biomass during several days
(Ridame et al., 2014; Guieu et al., 2014b; Tsagaraki et al., 2017). In addition to being strongly
stimulated by atmospheric P (Ridame et al., 2013), the trace metals in dust deposition have been
also suspected to stimulate $N_2$ fixation in the Mediterranean Sea (Ridame et al., 2011). The
extension of this fertilizing effect of dust events over the Mediterranean has been pointed out
from statistically positive correlations between dust deposition and surface chlorophyll
concentrations from remote sensing and modelling approaches (Gallisai et al., 2014). A
negative effect of atmospheric deposition on chlorophyll is, however, observed in the regions
under a large influence of aerosols from European origin (Gallisai et al., 2014). Indeed, the
input of anthropogenic aerosols, as Cu-rich aerosol, has been suspected to inhibit phytoplankton
growth (Jordi et al., 2012). Besides phytoplankton, dust deposition modifies also the bacterial
community structure by selectively stimulating and inhibiting certain members of the bacterial
community (Pulido-Villena et al., 2014; Tsagarakis et al., 2017). The budgets established from
4 artificial seeding experiments during project DUNE (Guieu et al., 2014b) all showed that
stimulating predominantly heterotrophic bacteria, atmospheric dust deposition can enhance the
remineralization of dissolved organic carbon (DOC), thereby reducing net atmospheric $CO_2$
drawdown. This also reduces the fraction of DOC that can be mixed and exported to deep waters
during the winter mixing (Pulido-Villena et al., 2008). Similarly, dust addition using on-land
mesocosms in the eastern Mediterranean Sea suggested that the auto- and hetero-trophic
components of the food web were enhanced by the dust addition thanks to the nitrogen and



phosphorus added through dust (Pitta et al., 2017 and companion papers) and that the response
was independent of the way the dust was added to the surface waters (single strong pulse or
three repetitive smaller pulses). One of the most intriguing results is the role of Saharan dust
deposition in the export of particulate organic carbon (POC) to the deep Mediterranean Sea by
both fertilizing and acting as ballast and facilitating aggregation processes (i.e. Ternon et al.
2010, Bressac et al., 2014; Desboeufs et al., 2014; Louis et al., 2017; Guieu et al., in prep.).
Experimental approaches have shown that wet dust deposition events, by supplying
bioavailable new nutrients, presents a higher positive impact compared to dry deposition, on
both marine primary production, nitrogen fixation and chlorophyll concentrations  (Ridame et
al., 2014; Guieu et al., 2014b).
Over the past decade, most of these valuable findings have been made thanks to experimental
approaches based on dust and aerosols addition into bottles and up to large in-situ mesocosms
or using remote sensing approaches. In this paper, we provide a state of the art regarding dust
deposition and its impact in the Mediterranean Sea and we describe our strategy to go a step
forward by catching a real wet deposition event at sea in order to study in situ the effects of the
rapid introduction of chemical elements and particles from the atmosphere onto the marine
element cycles, the biology and the export of material to the deep waters.
**2.  PEACETIME objectives**
In this context, the PEACETIME project (ProcEss studies at the Air-sEa Interface after dust
deposition in the MEditerranean sea) (http://peacetime-project.org/; last access 9 Feb. 2020)
aimed at extensively studying and parameterizing the chain of processes occurring in the
Mediterranean Sea after atmospheric deposition, especially of Saharan dust, and to put them in
perspective of on-going environmental changes. The ultimate goal was to assess how these



mechanisms impact, and will impact in the future, the functioning of the marine biogeochemical
cycles, the pelagic ecosystem and the feedback to the atmosphere.
The PEACETIME project was centered on a one-month oceanographic cruise in the central and
western Mediterranean Sea in May-June 2017. The strategy during the cruise was designed to
tackle the following questions:

1. How does atmospheric deposition impact trace element distribution in the column

water including the sea surface microlayer?

2. What is the role of dissolved organic matter/particulate dynamics on the fate of

deposited atmospheric trace elements?

3. How does atmospheric deposition impact biogeochemical processes and fluxes? Do in

situ biogeochemical /physical conditions matter?

4. What is the impact of atmospheric deposition on biological activity and on the

structure and composition of the planktonic communities?

5. How does atmospheric deposition impact the downward POC export and the

subsequent carbon sequestration?

6. What is the impact of biogeochemical conditions on gases and aerosol emissions from

the surface water?

7. How are optical properties above and below the air-sea interface impacted by aerosols

emission and dust deposition?

During the 33 day cruise, 40 scientists from the atmosphere and ocean communities travelled
2750 nautical miles (4300 km) performing simultaneously in situ sampling in the lower
atmosphere and the water column, and conducting on board experiments in climate reactors
simulating present and future marine physical conditions. The impacts on the cycles of chemical
elements, on marine biogeochemical processes and fluxes, on marine aerosols emission were




investigated in a variety of oligotrophic regimes. Characterizations of the chemical, biological,
physical and optical properties of both the atmosphere and the sea-surface microlayer, mixed
layer and deeper waters were performed.
The time of the campaign and the adaptive strategy for the cruise track, based on the daily
analysis of a number of operational forecast and near real-time observational products were
designed to maximize the probability to catch a Saharan dust deposition event in a stratified
water column in order to follow in-situ the associated processes. In this paper, we describe how
our strategy was designed before the campaign and how we were able to adapt it during the
cruise in order to sample a Saharan dust deposition event at sea, thanks to a dedicated and so-
called Fast Action prompted during the cruise.
**3.**    **Best time to schedule PEACETIME cruise**
In order to fulfil the objectives of the PEACETIME cruise, the occurrence probability of a
significant atmospheric deposition event was maximized by choosing to do the cruise during a
period of surface water stratification. This criterium matters because atmospheric inputs can be
the main external nutrient supply to offshore surface waters during the stratification period
(Guerzoni et al., 1999; The Mermex Group, 2011; Richon et al., 2018a). The Mediterranean
surface mixed layer depth monthly climatology (figure 1) shows a basin scale deepening from
November to February–March and an abrupt re-stratification in April, which is maintained
throughout summer and early autumn (D'Ortenzio et al., 2005). With mixed layer depths below
30 m in the whole Mediterranean basin, the May-September period looks particularly favorable
to sample highly stratified waters, with possible consideration of April and October months
(<40 m).



Because African dust transport associated to rain period generally leads to the highest
atmospheric deposition fluxes in the Mediterranean region (e.g., Loÿe-Pilot et al., 1986;
Kubilay et al., 2000; Fu et al., 2017), we checked the probability that a Saharan dust event may
occur during the cruise. The satellite-derived monthly climatologies of dust in the atmospheric
column over the Mediterranean show a maximum in summer in the western basin and in spring
and summer in the central basin (e.g., Moulin et al., 1998; Varga et al., 2014). Consistently,
model results in Figure 2 shows the highest values of dust aerosol optical depth (>0.10 and up
to 0.30 at 550 nm) over the whole western and central basins from May to August, an
intermediate situation in April and September, and the lowest values (generally <0.10) from
October to February. In addition to this seasonality of the dust columnar load, the climatology
of $PM_{10}$ and associated African dust concentration at the surface in the Mediterranean indicates
that the occurrence of dust plumes close to the surface, i.e. prone to dry deposition, is maximum
in April-May in Greece, April-June in Sicily, May-June in continental Italy, May in SE France,
June-July in NE Spain and July-August in SE Spain (Pey et al., 2013). From weekly insoluble
deposition monitoring at 4 sites of western Mediterranean islands (Frioul, Corsica, Mallorca
and Lampedusa) in the period 2011-2013, Vincent et al. (2016) report that most of the most
intense African dust deposition events occurred between March and June.
Literature from deposition measurements at various sites in the western Mediterranean
highlights a spring maxima for dust deposition (Bergametti et al., 1989; Loÿe-Pilot and Martin,
1986; Avila et al., 1997; Ternon et al., 2010; Desboeufs et al., 2018). Moreover, observations
indicate that the highest deposition fluxes of dust are most often associated with wet deposition
episodes (e.g. Loÿe-Pilot et al., 1986; Bergametti et al., 1989; Guerzoni et al., 1995; Loÿe-Pilot
and Martin, 1996; Avila et al., 1997; Kubilay et al., 2000, Dulac et al., 2004; Guieu et al., 2009;
Ternon et al., 2010; Vincent et al., 2016). A survey of dust wet deposition events at Montseny



stations in NE Spain over 1996-2002 concluded that the maximum frequency was in May (about
3 events per month) and June and November (about 1 event per month). Data from Vincent et
al. (2016) also show that most of the two-three highest dust deposition events recorded at each
of the 4 island stations cited above occurred between March and May, and are most often
associated with rainfall.
It was also important that the cruise crossed different trophic regimes to get likely contrasted
responses to atmospheric deposition. Although the Mediterranean is classified as an
oligotrophic basin characterized by low-nutrient concentrations, there is a general west-to-east
gradient of increasing oligotrophy (The Mermex Group (2011) and references within). Figure
3 shows monthly averaged satellite-derived Chl-a concentrations in the Mediterranean basin :
from April to June, various trophic conditions can be found in the basin, with still relatively
"high" Chl-a concentrations (0.3 mg m$^{-3}$) in the Ligurian and Alboran Sea and ultra-oligotrophic
conditions in the central and eastern basin (< 0.03 mg m$^{-3}$) (Bosc et al., 2004).
From all the preceding considerations, we finally concluded that mid-April to mid-June was the
target period for the cruise.
**4.   Spatial consideration: transect of principle of the PEACETIME cruise.**
The central Mediterranean Sea (MS) was our main targeted area since all the marine ecoregions
of the MS can be found in a relatively small zone (figure 4). Each ecoregion detected on that
figure presents a characteristic species association from primary producers to top predators of
the epipelagic domain, forced by similar environmental conditions (Reygondeau et al., 2014).
As seen in figure 4, the initial transect designed for PEACETIME aimed at visiting most of the
identified ecoregions within the 4 weeks of cruise, allowing us to test the impact of atmospheric
deposition on a large range of natural assemblages. The planned long stations of the transect of





principal were located within or at the center of 3 main ecoregions. Short stations (occupation
time was less than 6 hours) were positioned in order that cruising between two stations was
long enough (~8 hours) to allow the continuous measurement of both lower atmosphere and
surface seawater while cruising. Depending on atmospheric conditions during the cruise, it was
anticipated that one of the long station (named FAST) would be dedicated to documenting a
strong deposition event at sea, and that the forecasted occurrence of such an event would prompt
a fast action plan that might lead to change the planned transect and ship route.
It has to be noted that the coastal climate observatory of the Italian National Agency for New
Technology, Energy and Environment (ENEA) on Lampedusa Island (35°31'06"N,
12°37'48"E; figure 5) could provide an excellent support for atmospheric conditions in the
central basin before and during the cruise. Indeed, continuous measurements of aerosols
concentrations (Marconi et al., 2014) and composition (Becagli et al., 2012 and 2013), nutrients
deposition (Galletti et al., this issue), dust deposition (Vincent et al., 2016), optical
measurements (Meloni et al., 2007) and the vertical distribution of aerosols in the atmospheric
column by lidar (e.g. Di Iorio et al., 2009) are conducted at this site. During the cruise, 15
AERONET stations (Holben et al., 1998) plotted in figure 5 also provided continuous daytime
measurements of the spectral aerosol optical depth (AOD).
**5.  Implementation of the PEACETIME cruise**
Based on the scientific arguments detailed above and on the availability of the ship, the
PEACETIME cruise was conducted during late spring conditions from May 10 to June 11, 2017
on board the R/V *Pourquoi Pas ?* Along the 4300 km transect, 10 short stations (with an average
duration of 8 hours) and 3 long stations (respectively 4 days, 4 days and 5 days duration) were
occupied (figure 5 and table 1).



Everyday, thanks to the PEACETIME Operation Center (POC; see next section) based on land,
the relevance to follow the initial track was discussed in the view of several types of available
or derived products from various operational centers producing model forecasts and near-real
time remote sensing products). Figure 5 represents both the planned and realized transect
following the day-to-day adaptive strategy.

5.1 **Tools for decision: the PEACETIME Operation Center**

Based on the experience of the ChArMEx airborne campaigns (Mallet et al., 20016) and of
previous oceanographic cruises needing an adaptive planning strategy based on observations
and short-term forecasts (see section "Satellite monitoring of the ocean"), an operational server
named the PEACETIME Operation Center (POC; http://poc.sedoo.fr/; last access 9 Feb. 2020)
was set-up by the Service de Données de l'Observatoire Midi-Pyrénées (OMP/SEDOO,
Toulouse, France) for the cruise. It operated from early May to mid-June 2017, gathering a set
of quick-looks of (i) near-real time selected remote sensing or other observational products and
(ii) meteorological and chemistry-transport model forecasts, considered useful for the campaign
planning decisions. The quick-looks were either directly transferred to the POC following their
production by respective operational centers, or linked from their original browser. Various
reports were also produced and made available on a quasi-daily basis (meteorology and dust
over the basin, regional and local oceanographic conditions (SPASSO; see hereafter), ship
trajectory...    The    complete    series    of    reports    is    available    at
http://poc.sedoo.fr/source/indexGarde.php?current=20170602&nav=Reports    (last  access  9
Feb. 2020).  In the following, more details will be given on products that were found the most
useful for daily decisions during the cruise.
The actual positions of stations were discussed and determined on the basis of near-real time
satellite data analysis (SPASSO, see later) in order to account for local oceanic conditions (i.e.



presence or not of mesoscale structures). In parallel, short- and middle-term forecast models of
weather conditions and of dust transport and deposition were systematically analyzed to verify
the conditions, and eventually start the Fast Action. The Fast Action strategy consisted in
routing the ship towards an area of forecasted dust deposition event in order to tentatively
document the respective roles of dynamics and deposition on marine biogeochemical
conditions. The goal was to position the ship in the center of the area of dust deposition, at least
one day (24 hours) before the event in order to sample the water column before, during and
after the deposition, and collect and characterize the rain event. Several constraints had to be
considered for the Fast Action decision:
1. the uncertainties of the operational forecast models, which increased proportionally to

the length of the forecasted period;

2. the relative position of the ship (which was following the initial plan) and the forecasted

area of deposition, accounting for the maximum cruise speed of the ship (i.e. 15 knots)

and the need to be positioned at the station 24 h before the deposition event;

3. national and international authorisations related to the EEZ (Exclusive Economic

Zones); the Mediterranean area is almost completely submitted to national EEZ of

surrounding countries and, consequently, international oceanic areas are very scarce;

authorisation to sample an EEZ should be demanded in advance to the corresponding

country (generally 24 h or 48 h before, depending on the country).

All these elements were simultaneously analysed during a daily meeting between scientists
involved on land and on ship, as well as with the crew. Each day, the initial plan was confirmed
for the next 48 h or, eventually, modified. For most of the cruise (see figure 5), only slight
modifications of the initial plan were decided, as atmospheric conditions were not considered
favorable for the Fast Action. They dramatically changed on the 28th of May, during the





sampling of the ION station, conducting to the decision to start the Fast Action. The sequence
of events leading to the Fast Action are described later.

**285     Atmospheric conditions**

Several near-real time remote sensing products and model forecasts were used. In terms of
aerosol remote sensing, we mainly relied on two products. The first one was the aerosol optical
depth at 550 nm ($AOD_{550}$) distribution over the sea, as produced in near-real time by the ICARE
data and service centre, Lille, France (product SEV_AER-OC-L2; http://www.icare.univ-
lille1.fr/projects/seviri-aerosols; last access 9 Feb., 2020). Data from the Spinning Enhanced
Visible and Infra-red Imager (SEVIRI) on-board the geostationary satellite Meteosat Second
Generation (MSG) are directly acquired every 15 min by the Service d'Archivage et de
Traitement Météorologique des Observations Satellitaires of the Centre de Météorologie
Spatiale (CMS/SATMOS), Lannion, France, and processed within hours by ICARE based on
the algorithm of Thieuleux et al. (2005). The MSG satellite position at 0° longitude allows a
good coverage for aerosol climatologies and case studies of aerosol transport over the
Mediterranean basin (e.g. figure I.19 in Lionello et al., 2012; Chazette et al., 2016 and 2019)
and surrounding continental regions (Carrer et al., 2014) as well as of desert dust source regions
in Africa (e.g. Gonzales and Briottet, 2017). In addition to the quick-look from the level-2
product (SEV_AER-OC-L2) available between 4:30 and 18:00 UT at the maximum in mid-
June in our area of interest, a daily mean level-3 (SEV_AER-OC-D3) is produced every night
by averaging all available time slots during the previous day between 4:00 and 19:45 UT. Figure
6 illustrates this product for the 3rd of June when an African dust plume from North Africa
associated to a cloudy air mass invaded the westernmost Mediterranean basin atmosphere. The
horizontal resolution of the product is of 3 x 3 $km^2$ at nadir, of the order of 12.5 $km^2$ in the
Alboran Sea, 15 $km^2$ in the North of the Gulf of Genova, and 18 $km^2$ in the northeasternmost



basin (about 13.07, 13.64, and 13.96 at the FAST, ION, and TYR station, respectively).
Although less accurate than AOD from MODIS when compared to AERONET data, the high
temporal resolution of MSG/SEVIRI-derived AOD offers a much better daily coverage of the
area than any orbiting satellite (Bréon et al., 2011), especially when partial cloud coverage can
be compensated thanks to successive images, as illustrated in figure 6.
The second useful remote sensing product was the North African Sand Storm Survey
(NASCube) also produced from MSG/SEVIRI, at the Laboratoire d'Optique Atmosphérique,
Lille, France (http://nascube.univ-lille1.fr; last access, 9 Feb. 2020). It generates continuous
day and night images of desert dust plumes over the northern African continent and Arabian
Peninsula, using an artificial neural network methodology producing colour composite images
by processing 8 visible, near-infrared and thermal infrared bands of SEVIRI (Gonzales and
Briottet, 2017). Figure 7 shows a window of this product for the 1$^{st}$ June 2017, showing the
probable dust source regions (south of Morocco and western Algeria) of the plume found the
following days over the westernmost Mediterranean basin as seen in figure 6.
During the campaign, we also used on a regular basis air mass trajectories computed with the
Hysplit tool of the Air Resources Laboratory of the National Ocean and Atmosphere
Administration (NOAA/ARL; https://ready.arl.noaa.gov/HYSPLIT_traj.php; last access 9 Feb.
2020; Stein et al., 2015; Rolph et al., 2017) based on global meteorological 192-h forecasts
from the Global Forecasting System (GFS) model (1-deg, 3-h resolution) operated by the
National Centers for Environmental Prediction (NCEP; Yang et al., 2006). It could be used both
in forward mode to forecast the transport over the western Mediterranean of dust plumes
detected over Africa by NASCube, and in backward mode to identify the origin of air masses
over the ship position.



In addition to aerosol remote sensing observations we also used near real time rainfall remote
sensing    produced    by    the    Meteo    Company,    an    international    weather    network
(https://meteoradar.co.uk; last access 9 Feb. 2020) providing every 15 mn real time weather
radar- and satellite-derived maps of precipitation, clouds, and lightning on a European window
covering most of the Mediterranean basin (north of  32°N or 35.5°N, depending on products).
The satellite infrared images from SEVIRI are filtered to show the thicker clouds, and
observations from 45 European rain radar are integrated. Figure 8 illustrates the combined
SEVIRI satellite and radar product showing both clouds, precipitation and lightning for two
time slots on 3 June 2017. They show the beginning and the end, respectively, of a convective
rainfall of low intensity (<2 mm h-1) between Algeria and Spain in the dusty and cloudy area
visible in Figure 6 west of the ship.
A number of operational forecast models were also used, both for weather forecast and aerosol
transport. In order to understand the synoptic circulation, we especially considered surface
pressure (*P*) and 500-hPa (about 5.5-km altitude) geopotential (*Z500*) maps over the European
domain covering the whole Mediterranean basin and northern Atlantic from the global
numerical weather prediction model ARPEGE  (Courtier and Geleyn, 1988), developed and
maintained at Météo-France. Its horizontal resolution varies from 7.5 km in France to 37 km
over Southern Pacific, and four daily forecasts including data assimilation are carried out every
day (available  by http://www.meteociel.fr, last access 9 Feb. 2020). Because we were
especially targeting possible aerosol deposition events, we also analysed daily a set of up to 5-
days, 1-, 3-, or 6-hourly depending on models, precipitation forecasts from several models
including those made available by meteociel.fr including global weather forecast models such
as ARPEGE, IFS (the model developed at ECMWF; Barros et al. 1995), the Canadian CMC-
MRB GEM model (Côté and Gravel, 1998), the GFS atmospheric model from NCEP





(Kanamitsu, 1989) and its ensemble GEFS, but also the regional non-hydrostatic model
AROME (Seity et al., 2011) for the NW Mediterranean only at 1.3 km resolution.
Three regional dust transport models have also been considered, namely SKIRON operated by
the Atmospheric Modeling and Weather Forecasting Group (AM&WFG) of the University of
Athens (Kallos et al., 2009; Spyrou et al., 2010) and the two models NMMB-BSC (Non-
Hydrostatic Multiscale Model; Pérez et al., 2011) and BSC-DREAM8b (Basart et et al., 2012)
operated by the Barcelona Supercomputing Centre (BSC). SKIRON and BSC-DREAM8b have
a horizontal resolution of 0.24° and 0.33°, respectively, and are both initialized and constrained
at their boundaries by NCEP/GFS 6-hourly data. NMMB-BSC regional model has a resolution
of 0.47° x 1/3° and is constrained by the NCEP global version of the model (Pérez et al., 2011).
In terms of dust transport modeling, we mainly relied on 6-hourly dust optical depth and dry
and wet dust deposition fluxes forecasted daily from 12 UTC over the next 72 h by the NMMB-
BSC-Dust and BSC-DREAM8b v2.0 models and over the next 180 h (5.5 d) by SKIRON.
Because of its longer temporal range of forecast, the wet dust deposition product by SKIRON
was particularly useful to issue an early pre-alert for the Fast Action during the cruise. Figure
9 compares the forecast maps of 6-h accumulated dust deposition flux at 4 time steps from 3[rd]
June 2017 12 UTC to 5 June 00 UTC, from the 2nd June runs of those 3 models. This period
corresponds to the scavenging of the dust plume shown in Figure 6 that was targeted for the
Fast Action (see below).
We also used a set of forecast of aerosol or dust optical depth from a series of models: (i) 60-h,
6-hourly ensemble and comparative forecasts of dust optical depth from models operated by
the BSC for the World Meteorological Organization (WMO) Sand and Dust Storm Warning
Advisory and Assessment System (SDS-WAS), and made available by the Spanish Agencia
Estatal de Meteorologia (AEMET; https://sds-was.aemet.es/forecast-products/dust-forecasts/;





last access 9 Feb. 2020; it is worth noting that Basart et al. (2016) model data comparison over
summer 2012 showed better average performances of the model ensemble dust forecasts
compared to forecasts from any individual model (ii) 5-days, 3-hourly dust and sulphate AOD
Copernicus/GMES products over Europe and North Africa produced by the European Center
for Medium-Range Forecast (ECMWF), and (iii) 114-h, 6-hourly sulfate, dust and smoke AOD
over Europe and the Mediterranean region north of 35°N from the Naval Research Laboratory
(NRL) global NRL Aerosol Analysis and Prediction System (NAAPS) model that is using an
AOD assimilation package (Zhang et al., 2008); further, we used the kml formatted animations
of the NAAPS 5-days global forecasts of AOD projected on a GoogleEarth satellite view
centered on the western Mediterranean, which shows areas with significant AOD (>0.1) of
sulfate, dust or smoke. Finally, we also considered the daily maps (at time 00 UTC) produced
by the Earth Wind Map community (https://earth.nullschool.net; last access 9 Feb. 2020),
consisting of AOD from sulfate or dust from the NASA Global Modeling and Assimilation
Office (GMAO) Goddard Earth Observing System version 5 (GEOS-5) model overlaid by
surface or 700 hPa winds from the GFS model in order to check the dominant aerosol type and
transport conditions at the ship position.
**Ocean conditions**
Concerning the surface ocean, several remote-sensing datasets were exploited using the
SPASSO (Software Package for an Adaptive Satellite-based Sampling for Ocean campaigns
https://spasso.mio.osupytheas.fr/; last access: 9 Feb. 2020) in order to guide the cruise through
a Lagrangian adaptive sampling-strategy aiming at avoiding region of complex circulation and
dynamics (fronts, small scale eddies). The idea behind this approach was to aim at a situation
where the air-sea exchanges dominate and lateral advection and diffusion can be neglected.
Such an approach was already successfully adopted during several previous cruises such as



LATEX (Nencioli et al., 2011; Doglioli et al., 2013, Petrenko et al., 2017), KEOPS2 (d'Ovidio
et al., 2015), OUTPACE (Moutin et al., 2017; de Verneil et al., 2018) and OSCAHR (Rousselet
et al., 2019). During PEACETIME, we used the following datasets: (1) altimetry data from the
AVISO Mediterranean regional product (https://www.aviso.altimetry.fr/data/products/sea-
surface-height-products/regional/mediterranean-sea-gridded-sea-level-heights-and-derived-
variables.html); the altimetry-derived currents were then processed by SPASSO to derive
Eulerian and Lagrangian diagnostics of ocean circulation: Okubo-Weiss parameter, particle
retention time and advection, Finite Size Lyapunov Exposant (e.g. figure 10); (2) the sea surface
temperature (level 3 with resolutions of 4 and 1 km) and (3) the chlorophyll concentration (level
3 with a resolution of 1 km, MODIS Aqua and NPPVIIRS sensors combined after May 27,
2017 into a unique product) provided by CMEMS - Copernicus Marine Environment
Monitoring Service (http://marine.copernicus.eu/).
**5.2 The Fast Action**
**The decision process and atmospheric conditions**
On the 28$^{th}$ of May, the ship was ending the long station ION in the Ionian Sea under a
continuing northern atmospheric flux. A low pressure system reaching Spain from the Atlantic,
a typical situation for African dust transport in summer in this area (Moulin et al., 1998) caused
a southern flux over the western basin. But no significant emission of dust was yet detected in
Africa with NASCube. Aerosol transport models forecasted, however, the presence of dust
plume of moderate intensity for the following days, mainly confined to the southern part of the
western Mediterranean basin following a persistent western flux for several days limiting the
extension of dust transport towards the north of the basin. Some dust was predicted by NMMB-
BSC and SKIRON model runs of 27 June to be deposited by rain south of the Balearic islands
on 30 and 31 May, but meteorological forecasts did not converge on the time and location of





precipitation. In addition, the possible area of dust deposition was far from the ship, 16° in
longitude west from the ION station. Consequently, no modification of the initial plan was
decided and the station 8 was carried out southwest of Sicily on 30 May.
At the end of station 8 on 30[th] May, satellite observations confirmed the presence of atmospheric
dust in a cloudy air mass over the western part of the Mediterranean and long-term predictions
of AOD indicated the continuing presence of dust over the Alboran Sea, with a new dust plume
likely extending northwest on June 2 or 3. Although models still diverged in forecasting rain
over this region, the southwestern part of the Mediterranean basin looked to be the most dusty
area for the next days and it was decided to modify the initial plan and to move towards the
west for the last part of the cruise (see figure 5, the long transect south of Sicily).
The 31[th] of May the ship reached a position approximately located between Sicily and Sardinia
islands. Significant dust emissions were again observed over North Africa from the night of 30-
31 May on, and the predictions for a new significant dust event over the southwestern
Mediterranean on June 3-5 were confirmed. Although the differences between the models were
still important (only SKIRON forecasted a wet deposition event south of Spain for the 3-4[th] of
June), it was decided to continuously move the ship westward, and to shift station 9 from its
initial position in the Tyrrhenian Sea to a new position in the Alboran Sea. We considered that
another station in the Tyrrhenian area was not critical for the cruise objectives and that
establishing the area of next operations in the Alboran Sea could facilitate the re-positioning of
the ship in the case of a confirmed prediction of a wet deposition event.
The 1[st] of June, during the sampling at station 9 midway between Sicily and Spain, it was
decided to start the Fast Action. Indeed, dust emissions continued in Algeria and southern
Morocco associated to a southern flux, aerosol transport models confirmed a new significant
dust episode with AOD >0.8 (i.e. roughly 1 g m$^{-2}$ of dust in the atmospheric column) for June



3-5, and the occurrence of associated rains appeared most likely from most meteorological
forecasts. SKIRON and NMMB-BSC predicted the dust wet deposition flux to be more
important on 3rd June in the Alboran Sea west of 0° longitude, of the order of 1.5 and 0.5 g m-
2, respectively), but longer-term forecasts by SKIRON predicted wet dust deposition more east
south of the Balearic Islands on June 4 (~0.5 g m$^{-2}$) and especially on first half of June 5
(possibly >1.5 g m$^{-2}$), a possibility confirmed by other rain forecasts. The Fast Action station
was positioned 145 km south the Balearic Island of Mallorca and 126 km north of the Algerian
coast (Figure 5), where a limited portion of the sea is part of international waters (i.e. not
included in an EEZ), and in an area where the influence of Atlantic waters reacher in nutrients
than Mediterranean waters should be limited compared to the more western Alboran Sea. The
ship reached the FAST station location on the 2nd of June at the end of the day (Table 1) and
the ocean and atmospheric sampling started immediately.
Although cloudy, only from the 3rd of June rain conditions were observed in the neighbouring
area (see rain radar composite images in figure 11). The SEVIRI AOD remote sensing
confirmed the export of a dust plume from North Africa south of the Balearic Islands with high
AOD (>0.8; Figure 6) and NASCube confirmed new dust emissions in the night from 3 to 4
June. The dust plume was transported to the NE up to Sardinia on June 4, with AOD <0.5 in all
the area and clear sky with low AOD was left west of 4°E on June 5 (Desboeufs et al., in
preparation, this issue). Wet deposition of dust for the 4th and early 5th June in the FAST station
area were confirmed by the deposition maps from the 3 regional dust transport model forecast
runs of June 2, although with decreasing intensity compared to the previous runs (except for
BSC-DREAM8b that did not forecast dust wet deposition in earlier run), from a very small flux
of a few mg m$^{-2}$ (BSC-DREAM8b) to about 100 mg m$^{-2}$ (NMMB-BSC) and up to more than 1
g m$^{-2}$ (SKIRON) (Figure 9).  A rain front, moving eastward from Spain and North Africa



regions, reached the Fast Action position the night between the 4th and the 5th of June (Figure
11). A single event of rain was observed and sampled on the ship at the station FAST on 5th
June from 2:36 am to 3:04 am. Continuous lidar measurements on board the ship confirmed the
presence of a dust layer mainly over the atmospheric boundary layer over the FAST station and
its below-cloud deposition during the rain event of early 5th June (Desboeufs et al., in
preparation, this issue). The chemical composition of this rain sample confirmed wet deposition
of dust reaching a total particulate flux of 12 mg.m$^{-2}$ (Fu et al., in preparation, this issue), which
is among the lowest most intense dust deposition fluxes recorded in this area from long time-
series of deposition network (Vincent et al., 2016).
**Sea Surface dynamic context at FAST**
Several approaches have been implemented to highlight the dynamical context around the
FAST station in the waters above 200m, the physical structures and the possible influences of
the dynamics on the stability of the water masses at the station. These approaches are based on
in situ observations (Moving Vessel Profiler (VMP) transect and drifters trajectories) and
diagnostic tools.
On board, a  MVP collected high frequency Conductivity- Temperature- Depth (CTD) data
along two transects: the first one when the vessel approached the FAST station from the east,
before the station took place, and the second one west of the FAST station, on the transect back
from station 10. Figure 12 shows these data in a longitude-depth section. To the east of the
FAST station the surface water was colder and saltier than to the west, where a strong
deformation of the isopycnals suggests the presence of an Algerian anticyclonic eddy. Such
eddy carries recent Atlantic water and generates a southward current that only partially impacts
the FAST station area.



The post-cruise comparison of the hull-mounted ADCP data combined with the SVP drifters
trajectories and the altimery-derived currents shows a good agreement all along the cruise and
in particular at the FAST station (Figure 12). Moreover, the agreement between SVP and
numerical particle trajectories has been slightly improved when we also took into account the
Ekman drift calculated with wind data from the high resolution regional model WRF 3.7.
This allowed us to calculate backward trajectories of the surface water masses using the
ARIANE Lagrangian tool (Blanke and Raynaud, 1997; Blanke et al., 1999) in order to estimate
the origins of the sampled surface water at the LD stations. As seen from the repeated CTDs
(see previous section), at the FAST station a southward current associated to the large Algerian
eddy was present. We estimated that over the whole station duration, a mean value of 57(26)%
of water remained in the station zone after 1(2) day(s). Moreover, combining the particle
trajectories and the precipitation data from the WRF 3.7 model we concluded that the rain,
which fell slightly upstream the LD-FAST station in the previous days, likely impacted the
sampled water mass (figure 14).

## Temporal evolution of surface seawater properties during FAST

Station FAST has been documented at its fixed point during seven days by 43 repeated CTD
casts in the depth range 0-200 m.
The hydrological situation was characterized by a very shallow surface mixed layer and a sharp
seasonal thermocline that extended underneath down to 75 m depth (Figure 15, upper right and
middle right panels). In this upper layer, salinity values were lower than 37.5, which is
characteristic of modified Atlantic waters flowing eastward inside the Mediterranean Sea. In
the deeper layers, salinity increased sharply with depth until 350 m where it reached its
maximum value (38.59), which is characteristic of Levantine intermediate waters flowing





westward into the Mediterranean outflow. Deep waters, formed at winter convection zones of
the northwestern Mediterranean, had lower salinity values (38.48); they extended from 1400 m
down to the sea bottom.
The hydrological conditions at this site between June 2 and 8 during the Fast Action mainly
evolved in the upper layer (Figure 15, upper left and middle left panels). The surface mixed
layer was shallow with variations from 9-m to 19-m depth following the diurnal cycle. Mixed
layer salinity remained equal until the 7$^{th}$ of June; in particular no dilution effect due the rainfall
on 3$^{rd}$ June has been recorded. The stratification of the whole water column remained steady
during the long station. Density horizons kept lain along isobars in the upper layer, which signs
the absence of geostrophic perturbations during the long station. However, the current profilers
indicated a depth-independent (barotropic) motion of amplitude 3 cm s$^{-1}$ heading 220°, which
is in agreement with the position of the station within the large eastern Algerian Gyre, a
component of the basin scale cyclonic circulation described by Testor et al. (2005). This
southwestward flow transported superficial water masses of distinct properties as clearly
marked below the mixed layer by salinity anomalies (referenced to the initial profile of 2$^{nd}$ June
16:30). These water masses crossed the observation site, disrupting the water column in the
depth range of 25-100 m, lowering salinity values by 0.1 in the extension of the thermocline
and increasing salinity values by 0.05 underneath.  Although clearly present, this hydrological
anomaly did not affect the surface waters and the MLD was stable during the Fast Action
precluding any input from below that could have been linked to destratification induced by
strong wind associated to the dust event as hypothesized in Guieu et al. (2010) from time-series
observations in the northwestern Mediterranean Sea. Such conditions are favorable to observe
any change strictly attributed to external inputs from above (i.e. atmospheric deposition).

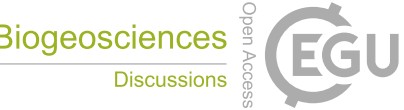

The distribution of phytoplanktonic biomass has been detected by optical sensors mounted in
the CTD package (Figure 15 lower panels): measurements of fluorescence and of beam
transmission provided similar patterns, stressing the biogenic character of particles present in
the water column. Intermittent signal at the sea surface has been detected only by
transmissometry, however no clear relationship with the rainfall event can be stated (see the
first profile after event in red). The vertical distribution was displayed as a deep chlorophyll
maximum of about 20-m thickness, located at the base of the thermocline (about 75 m). Short-
term evolution during the seven days of observation displayed variations in intensity and depth
of the deep chlorophyll maximum, as well as splitting and merging sequences of the peak. Such
perturbations appeared after the rainfall event of the 3$^{rd}$ June, however they more likely result
from the intrusion of water masses from north at this depth range. This hypothesis is reinforced
by the absence of any geostrophic perturbation in the density time series that could have injected
biomass or nutrients via diapycnal processes. Another candidate could be the mixing effect
associated to the breaking of internal gravity waves that propagated along the thermocline.

5.3 Work at stations and work underway

In the figure 16 we show the satellite-derived SST data averaged taking into account the ship
position. During the cruise a general warming of the sea surface is observed and the FAST
station has been performed in waters warmer than the two others LD stations.
Chlorophyll concentrations as seen by satellite over the western and central Mediterranean Sea
were typical of the oligotrophic conditions encountered during the season characterized by a
strong stratification (Figure 17). The west-east gradient between oligotrophic to very
oligotrophic was clearly established and minimal concentrations (about 0.05 mg m$^{-3}$ ) were
observed in the Ionian Sea.





Surface inorganic nutrients measured at nanomolar concentrations were very low for both
dissolved inorganic nitrogen (DIN) and phosphorus (DIP). Indeed, average concentration in 0-
20 m  layer was 90 nM DIN and 15 nM DIP at the westernmost station (station 10) and 14 nM
DIN and 10 nM DIP at the easternmost station (ION). Along the longitudinal transect, a
deepening of the nutrient depleted layer toward the east was observed (figure 18) consistent
with the general trend of those nutrients in the Mediterranean basin as described in Mermex
Group (2011) and references inside.
All along the cruise, the work at sea was divided between short (~8 hours) and long (up to five
days) stations to allow both a good description of the different ecoregions crossed and to
perform process studies. The number of short stations was the best compromise in order to (1)
allow at least 8 hours of transect at 9 knots in between 2 short stations, necessary for a good
continuous monitoring of both low atmosphere and surface waters while cruising and (2) to
have enough short stations to describe well enough stocks and fluxes along the whole column
water and microstructure of the mixed layer in the contrasted biogeochemical regions crossed.
Long stations were located in 3 different ecoregions (see Figure 3) characterized by different *in*
*situ* conditions (see figures 16 & 17) all characterized by oligotrophic conditions. The duration
of the long stations (4 days at TYR and ION and 5 days at FAST; table 1) allowed process
studies both in situ (drifting mooring supporting different types of traps and instruments) and
on board (artificial dust seeding experiment in 300 L climate reactors; see below). Table 2
summarizes the operations conducted during the cruise and the parameters obtained, (1) on a
continuous basis, (2) at short stations, and (3) at long stations .  A summary of the use of those
parameters in the different papers presented in this Special Issue and in other papers is also
given in that table.





### 6. Conclusion
The PEACETIME oceanographic expedition conducted in spring 2017 cruised over a 20°
longitudinal gradient across the western and central Mediterranean Sea during the season
characterized by strong stratification, low productivity and high chance to be submitted to dust
wet deposition. Those conditions were required in order to fulfil the objectives of the project
aiming at quantifying the biogeochemical processes at play after atmospheric deposition and its
impact on ecosystem functioning. Thanks to an adaptive strategy based on a large panel of
atmosphere and ocean real time observations and forecast models, the track of the cruise was
optimized from day to day. In particular, we were successful to timely reroute the R/V toward
an area where dust deposition was expected and actually observed and sampled. Different
atmospheric situations were encountered during the cruise,  allowing the acquisition of a large
dataset under different dynamical and biogeochemical in situ conditions to explore the chemical
and ecosystem in situ response to deposition.
### Acknowledgments
This study is a contribution to the PEACETIME project (http://peacetime-project.org), a joint
initiative of the MERMEX and ChArMEx components supported by CNRS-INSU, IFREMER,
CEA, and Météo-France as part of the programme MISTRALS coordinated by INSU.
PEACETIME was endorsed as a process study by GEOTRACES. PEACETIME cruise
https://doi.org/10.17600/17000300. We thank the captain and the crew of the RV *Pourquoi Pas*
*?* for their professionalism and their work at sea. We warmly thank Louise Rousselet, Alain de
Verneil, and Alice Della Penna for their precious help with SPASSO and the daily bulletins,
Hélène Ferré from OMP/SEDOO for her management of the Peacetime Operation Center, and
Louis Prieur for fruitful discussion on the characterisation of the ocean processes observed at



the station FAST. We finally acknowledge the technical help of the many teams supporting
operational production of model forecasts and remote sensing products that contributed to the
POC and were the basis of our daily briefings and decision for the Fast Action, especially
Jacques Descloitres for MSG-derived AOD, Louis Gonzalez for NASCube products, and
Christos Spyrou for SKIRON forecasts.
**Author contribution:** CG and KD designed the PEACETIME project. FD and FDO analyzed
the FAST Action. FD, MM and PN analysed the atmospheric components and VT, AD, AP and
SB analysed the marine components. CG prepared the manuscript with contributions from all
co-authors.

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





**Table 1.** Date of occupation, position and depth of the short stations (ST1-ST10), of the long
stations (TYRR, ION, FAST) and of the SAV station.

|  | arrival date | local time | departure date | local time | depth m | lat N | long E |
|---|---|---|---|---|---|---|---|
| ST1 | 12/05/2017 | 05:45 | 12/05/2017 | 21:15 | 1580 | 41°53.5 | 6°20 |
| ST2 | 13/05/2017 | 06:30 | 13/05/2017 | 13:08 | 2830 | 40°30.36 | 6°43.78 |
| ST3 | 14/05/2017 | 06:00 | 14/05/2017 | 13:30 | 1404 | 39°08.0 | 7°41.0 |
| ST4 | 15/05/2017 | 05:56 | 15/05/2017 | 13:04 | 2770 | 37°59.0 | 7°58.6 |
| ST5 | 16/05/2017 | 04:00 | 16/05/2017 | 10:58 | 2366 | 38°57.2 | 11°1.4 |
| TYRR | 17/05/2017 | 05:08 | 21/05/2017 | 15:59 | 3395 | 39°20.4 | 12°35.56 |
| ST6 | 22/05/2017 | 04:50 | 22/05/2017 | 10:38 | 2275 | 38°48.47 | 14°29.97 |
| SAV | 23/05/2017 | 11:30 | 23/05/2017 | 14:17 | 2945 | 37°50.4 | 17°36.4 |
| ST7 | 23/05/2017 | 21:10 | 24/05/2017 | 07:15 | 3627 | 36°39.5 | 18°09.3 |
| ION | 24/05/2017 | 18:02 | 29/05/2017 | 08:25 | 3054 | 35°29.1 | 19°47.77 |
| ST8 | 30/05/2017 | 03:53 | 30/05/2017 | 09:41 | 3314 | 36°12.6 | 16°37.5 |
| ST9 | 01/06/2017 | 19:13 | 02/06/2017 | 04:41 | 2837 | 38°08.1 | 5°50.5 |
| FAST | 02/06/2017 | 20:24 | 07/06/2017 | 23:25 | 2775 | 37°56.8 | 2°54.6 |
| ST10 | 08/06/2017 | 05:12 | 08/06/2017 | 10:25 | 2770 | 37°27.58 | 1°34.0 |
| FAST -bis | 08/06/2017 | 21:06 | 09/06/2017 | 00:16 | 2775 | 37°56.8 | 2°55.0 |







**Table 2. Work at sea during the PEACETIME Cruise and associated publications**

| total number | Operations at sea | details | 1 | 2 | 3 | 4 | 5 | 6 | 7 | 8 | 9 | 10 | SAV | TYR | ION | FAST | Papers presenting the data |
|---|---|---|---|---|---|---|---|---|---|---|---|---|---|---|---|---|---|
| | | | SHORT STATIONS | | | | | | | | | | | LONG STATIONS | | | |
| continuous | Atmospheric sampling | [1] | CONTINUOUS | | | | | | | | | | | CONTINUOUS | | | Desboeufs et al., in prep (a), Desboeufs et al., in prep (b); Fu et al., in prep. (a); Garcia-Nieto et al., in prep.; Riffault et al., in prep. |
| 3 | Rain water collection | [2] | | | | | | | | | | | | | | x | x | Fu et al., in prep (b) |
| continuous | Continuous surface seawater pumping (-5 m) | [3] | CONTINUOUS | | | | | | | | | | | CONTINUOUS | | | Freney et al., in prep.; Sellegri et al., submitted; Trueblood et al., in prep. |
| 3 times | Surface seawater pumping (large volume (1800 L) for experiments in Climate Reactors) | [4] | | | | | | | | | | | | x | x | x | Gazeau et al., in prep (a & b); Ridame et al., in prep.; Roy-Barman et al., in prep.; Dinasquet et al., in prep.; Guieu et al., in prep. |
| 90 casts | Classical Rosette with 24 Niskin bottles or 22 Niskin bottles + 3 HP bottles [1]. | [5] | 0-500 and 0-bottom | | | | | | | | | | | | | | Taillandier et al., submitted; van Wambeke et al., in prep.; Maranon et al., in prep.; Berline et al., in prep.; Jacquet et al., in prep.; Zancker et al., in prep.; Barbieux et al., in prep.; Baumas et al., in prep. Guieu et al., this paper ; Garel et al., 2019 |
| 27 casts | Trace metal Clean Rosette on kevlar cable with 24 teflon-coated GoFlo bottles [2]. | [6] | 0-bottom | | | | | | | | | | | | | | Bressac et al., in prep.; Pulido-Villena et al., in prep.; Desgranges et al., in prep., Ridame et al., in prep. |
| 17 sampling | Microlayer sampling (rubber boat) | [7] | x | | | | | | | | | | x | xx | xx | xxxx | Tovar-Sanchez et al., in rev.; Zancker et al., in prep.; Engel et al., in prep. |
| 17 free-fall profiles | Optical measurements: HyperPro [3] | [8] | | x | x | x | x | x | x | | | | x | xxxx | xx | xxxx | |
| 23 net tows | Zooplankton Net (0-200 m) | [9] | x | x | x | x | x | x | x | x | x | | | xxxx | xxxxxx | xxxxxx | Feliu et al., in prep. |
| 13 | Marine snow catcher (depth m) | [10] | | | | | | | | | | | | 70 -80 -90 - 200 | 80 -100 - 150 - 200 | 70 -75 - 80- 100 | |





| Frequency | Instrument | Ref | | | | Reference |
|---|---|---|---|---|---|---|
| 3 times | Drifting mooring | [11] | x | x | x | Bressac et al., (2019); Bressac et al., in prep. |
| 3 times | Sediment Cores | [12] | x | x | x | Brandt et al., submitted; Brandt et al., in prep. |
| 20 drifters | SVP (Surface Velocity Program) drifters | [13] | x | x | | Menna et al., 2019. Guieu et al., this paper; Berline et al.in prep., Desboeufs et al., in prep. (b) |
| a total of 1000 profiles (0-300m) | Moving Vessel Profiler (begining and end long stations) | [14] | between the short stations and in the long stations area as frequent as possible | | | Guieu et al., this paper; Berline et al. in prep. |
| 2 deployments, 1 recovery | Biogeochemical ARGO float | [15] | | x | | Barbieux et al., in prep. ; Taillandier et al., submitted |

1] Atmospheric sampling was carried out throughout the transect using PEGASUS dedicated mobile platform (PortablE Gas and Aerosol Sampling UnitS, Formenti et al., 2019) to monitor continuous air gaseous composition (NOx, SO₂, O₃, CO₂, CO, VOC), physico-chemical properties of aerosol particles (mass and number concentration, size-distribution, chemical composition and nutrients contents), parameters of atmospheric dynamics such as the boundary layer, and radiative parameters (incident radiation, optical thickness, optical properties of the particles).

[2] Rain sampling of two events that occurred during the cruise. The on-line filtration collector was used to determine the dissolved and particulate composition of rain, including major and trace metals (Al, Ba, Cd, Co, Cr, Cu, Fe, Mo, Mn, Ni, Pb, Sr, Ti, V, Zn), atmospheric inorganic compounds (sulfate, chlorure, Na, Mg, K, Ca… ) and dissolved nutrients (phosphate, nitrate, ammonium).

[3] An innovative system of continuous "clean" pumping activated by a large peristaltic pump connected to a tube plunged at 5 m under the surface seawater inside a TraOcean was set up. The water was conveyed in a dedicated laboratory and distributed to several instruments to assess its



chemical properties (carbonate chemistry, $O_2$), microbial assemblages, hydrological properties, optical properties related to community and particle
composition and aerosol production (chemical composition, particles spectrum) throughout the transect.
[4] Climate reactor experiments: 6 large volume tanks (300 L) were filled with surface water. After artificial dust seeding at the surface of 4 of the
reactors, the impact of dust deposition on biogeochemical stocks and fluxes under present and future environmental conditions (acidification and
increase of the temperature of the sea water) was followed during 4 days (TYR and ION) and 5 days (FAST). [listing all parameters measured: see
Table 1, Gazeau et al., this issue (a)].
[5] The "classical" Rosette was composed of a CTD underwater unit that continuously collected the following parameters: pressure, temperature
and salinity of seawater, dissolved oxygen concentration, photosynthetically active radiation (PAR), beam transmission (at 650 nm), chlorophyll-
a fluorescence. A LISST (Laser *in situ* Scattering and Transmissiometry Deep (LISST-Deep), Sequoia Sc) was mounted independently on the CTD
frame. This instrumental package was also composed of a sampling system : 24 12-L Niskin bottles could be fired at specified levels during upcasts.
Some Niskin could be replaced by High Pressure (HP) bottles that allowed hyperbaric sampling on dedicated deep casts. Water from the classical
Rosette was used to quantify $O_2$, AT/CT, nutrients, DOC, POC/PON, hyperspectral particulate absorption coefficient, hyperspectral CDOM
absorption coefficient, chlorophyll pigments, viruses abundance and lysogeny, bacteria, flagellates and pico-nanoeukaryotes abundance (by
cytometry), total combined carbohydrates, total hydrolysable amino acids, gel particles (TEP and CSP), bacterial production, dissolved and
particulate primary production, virus diversity, and eukaryote diversity.



[6] The trace metal "clean" Rosette was composed of a titanium CTD underwater unit that continuously collected the following parameters:
pressure, temperature and salinity of seawater, dissolved oxygen concentration, CDOM Fluorescence. This instrumental package was also
composed of a teflon-coated sampling system: 24 GoFlo bottles could be fired at specified levels during upcast. Water from the clean rosette was
used to measure dissolved metals (Al, Cd, Co, Cu, Fe, Mo, Ni, Pb, V, Zn) and particulate (Al, Ba, Ca, Cu, Fe, Mn, Ni, Ti, Zn), total mercury,
methyl mercury, inorganic phosphate and nitrate (nano-molar), nutrients (to be measured with Technicon), di-nitrogen fixation, diazotrophs
diversity (only at Station 10).
[7] Discrete sampling of the surface micro layer (SML) was performed from a rubber boat using gas plate systems. Dissolved (<0.22 μm) and total
(unfiltered) SML samples were collected for trace metals (Cd, Co, Cu, Fe, Ni, Mo, V, Zn and Pb) and nutrients analysis. Same metals were also
measured in a subsurface (0-1 m) filtered (0.22 μm) sample. Samples were collected for the determination of total combined carbohydrates, total
hydrolysable amino acids, gel particles (TEP and CSP). DNA was extracted from filters from the surface microlayer and subsurface water (~ 20
cm). Three experimental SML additions were carried out in waters of TYR, ION and FAST stations.
[8] The HyperPro measured hyperspectral upwelling radiance (Lu) and downwelling irradiance (Ed) at the daily solar maximum.
[9] Samples collected by net hauls between 0 and 300 m performed with a BONGA net equipped with a 100 μm and a 200 μm mesh size.
Zooplankton abundance, biomass and taxonomy were obtained in 3 size classes: <200 μm; >200 - <1000 μm and >1000 μm. At long stations,
additional samples were taken for stable isotopes analyses.





[10] The large Marine Snow Catcher bottle (100 L) was used at long duration stations to collect suspended particles, slow sinking particles and fast
sinking particles. Heterotrophic production of prokaryotes attached to these different particles types was measured along with TEP abundance and
POC concentrations. In addition concentration kinetics of aminopeptidase, alkaline phosphatase and beta D glucosidase was measured. Diversity
of microorganisms collected in each type of particles was analyzed by barcoding and sequencing.
[11] The mooring was equipped with (i) 3 Technicap type PPS5 particle traps at 200, 500 and 1000 m, each equipped with inclinometers, (ii) 3
IODA (In Situ Oxygen Dynamics Auto-analyzer) at 5, 90 and 200 m; (iii) 2 in situ particle interceptor/incubator – RESPIRE (at 120 m and 200
m), (iv) 2 trace metal clean RESPIRE (at 110 m and 190 m), and (v) 1 Sediment Trap Station with 4, ⌀80-mm tubes in transparent PVC. The line
was also equipped with 4 CTD / O2 type SeaBird Microcat SBE37, 4 Aquadopp Doppler current meter from Nortek brand, 5 RBR Autonomous
Temperature and Pressure Sensors, 5 RBR Autonomous Temperature Sensors alone. Drifting moorings were deployed for 4 days at TYR and ION
and 5 days at FAST. Fluxes for particulate mass, carbon, organic carbon, inorganic carbon, nitrogen, calcium, aluminium, iron, biogenic and
lithogenic silica were determined from PPS5 samples.
[12] At TYR (depth 3395 m), ION (depth 3054 m) and FAST (2775 m), sediment core sampling were carried out with a multicorer, sliced into
depth layers to perform DNA extractions.
[13] A total of 20 SVP (Surface Velocity Program) drifters were deployed at the long duration stations to provide information on the current at 15-
m depth.



[14] A MVP (Moving Vessel Profiler) was deployed to perform high frequency 0-300 m profiles of CTD (and fluorescence and LOPC-Laser
Optical Particle Counter, when the "big fish" was towed instead of the "small fish") between the short stations and in the long station areas as
frequently as possible (see Figure 12). A total of more than 1000 profiles have been obtained.
[15] Two Biogeochemical Argo profiling floats have been deployed in the Ionian Sea. In addition to the CTD, the floats interfaced bio-optical
sensors that measured fluorescence of Chlorophyll and CDOM, particulate backscattering (700 nm), Photosynthetically Active Radiation and
downwelling irradiance at three wavelengths (380, 412, 490 nm). In addition, the float released at the ION station included an optode that measures
dissolved oxygen and a beam transmissometer (650 nm).






**Figure Captions.**
**Figure 1**. Mediterranean surface mixed layer depth (m) monthly climatology over 1940-2004
(in m; from D'Ortenzio et al., 2005; Copyright 2005 by the American Geophysical Union).
**Figure 2**. Monthly-averaged dust optical depth at 550 nm (1979-2013 period) over the
Mediterranean region from the CNRM-RCSM5 regional coupled climate system model (after
Nabat et al., 2015).
**Figure 3**. Monthly averaged chlorophyll maps derived from SeaWiFS data for the year 1999
(Bosc et al., 2004; Copyright 2004 by the American Geophysical Union).
**Figure 4.** Spatial distribution of the Mediterranean epipelagic marine ecosystems of the
Mediterranean Sea (from Reygondeau et al. 2014). The consensus regions (in white, from Ayata
et al., 2018) from eight regionalisations of the Mediterranean Sea, are characterised by well
defined, relatively homogeneous biogeochemical and hydrodynamical conditions, with similar
temporal dynamics). The transect initially planned is superimposed.
**Figure 5**. Transect of the PEACETIME Cruise: Initial (dotted line) and final track (continuous
line); stations are indicated by filled circles (planned stations: smaller, pink: realized: larger,
orange). The 10 short stations are numbered from St.1 to St.10. TYR, ION, and FAST indicate
the 3 long stations. The SAV station was only performed for the retrieval and launch of floats.
The land-based Lampedusa observatory (purple triangle) and 15 AERONET stations operated
during the cruise are also represented  (brown diamonds).
**Figure 6**. Aerosol optical depth at 550 nm derived from MSG/SEVIRI on 3 June 2017; left:
from the 15-mn image acquired at 13:00 UT; right: daily average from 52 images acquired
between 4 and 18:30 UT. The black circle indicates the position of the ship (station FAST) .





The dark grey mask corresponds to land and coastal ocean pixels, the light grey, to cloudy
pixels.
**Figure 7.** NASCube image window over North Africa and southern Europe for 1 June 2017,
02 UT. This nighttime image is derived from MSG/SEVIRI thermal infrared channels by
comparison to a clear reference image for the period, allowing detecting high dust load over the
continental surfaces (Legrand et al., 2001). White tones indicate clouds, the highest being the
brightest and the thermal anomalies attributable to dust are coloured by increasing intensity
from blue to pink. They are associated with increasing AOD from light blue (typically <0.3) to
purple (~1) and pink (>2) (Gonzalez and Briottet, 2017).
**Figure 8.** Rain-lightning-clouds (RLC) image window over the western Mediterranean and
Spanish Peninsula showing clouds (white areas), estimated precipitation (blue shades), and
lightning strikes (yellow circles) obtained by combining SEVIRI infrared images and European
rain radars (from meteoradar.co.uk; access 3 June 2017).
**Figure 9.** Maps of 6-h accumulated desert dust wet deposition fluxes in the western
Mediterranean produced by the forecast run of 2 June 2017 of the three dust transport models
NNMB-BSC-Dust-v2 (top) BSC-DREAM8b (middle) and SKIRON (bottom), at times 3 Jun,
12 UTC and 18 UTC, 4 Jun 18 UTC and 5 Jun 00 UTC from left to right, respectively.
**Figure 10.** Map of the FSLE (Finite Size Lyapunov Exposant, day$^{-1}$) calculated from the near-
real-time altimetry-derived surface currents for June 4, 2017. The figure is taken from the
SPASSO bulletin of June 5, 2017 with the planned stations shown in black and the route toward
the FAST station highlighted in magenta.
**Figure 11.** (1) Rain rate (mm h$^{-1}$) during the night between the 4$^{th}$ and 5$^{th}$ of June (white dot is
the position of the FAST station). These European radar composite products were provided by



the Odyssey system, created in the framework of the Opera Program that is the radar component
of the Eumetnet observation Program.
**Figure 12.** MVP measurements across the FAST station. In the upper panel, the positions of
each MVP cast (and of the FAST station) are shown as black (and red) crosses. Below are
shown the sections of temperature (top), salinity (middle) and density (bottom).
The map of the altimetry derived currents shows clearly the presence of the Algerian eddy west
to the FAST station sampled during the MVP transect (figure 13).
**Figure 13.** Geostrophic currents from satellite data with the Ekman component from WRF
model added (black arrows, mean during the shown transect). In addition, in situ drifter
trajectories during 30 days (launched at FAST and in its vicinity) are represented as white lines.
Horizontal currents measured by the VM-ADCP for the first two bins (purple arrows -18 m,
salmon arrows -26 m) are superimposed for comparison.
**Figure 14.** ARIANE particles initial positions (white) and after a backward integration of 1
(pink), 2 (light red), 3 (dark red), and 10 days (black) for the FAST station on the 3rd of June.
(a) large view, (b) zoomed view, (c) ratio of particles remaining in the initial zone as a function
of the number of backward integration days.
**Figure 15.** Left panels: temperature-salinity diagram (upper panel), temperature profiles
(middle panels), and profiles of beam transmission (lower panel). Right panels: evolution of
the surface stratification ($s_q$, upper panel), salinity (anomalies to the profile of $t_o - 2^{nd}$ June
16h30, middle panel) and chlorophyll fluorescence (lower panel) at the FAST station. Time
Series are composed of 43 repeated CTD casts, with variable temporal resolution. The depth
of the mixed layer is indicated by white dots. The time of the rainfall is indicated by the red
line.





**Figures 16.** Sea Surface Temperature during the cruise; left) outward route (10-28 May), right)
return route (28 May-10 June). The daily satellite pixel data are used to produce a weighted
mean. The weight for each pixel is calculated by normalizing by the square of the inverse
distance from the pixel to the daily mean ship position. The ship track is shown in black, the
short (long) station positions are indicated with black dots (squares). Data courtesy of
L.Rousselet.
**Figure 17.** As figure 17, but for the satellite-derived surface Chlorophyll-a concentration
averaged over the entire duration of the cruise. Data courtesy of L.Rousselet.
**Figure 18.** Nitrate (top right) and phosphate (bottom right; see Pulido-Villena et al., in prep.)
concentrations (in nM) above 100 m, during the PEACETIME cruise along the west-east
gradient shown on the map (left).
**Figures**

1300                                **FIG 1**

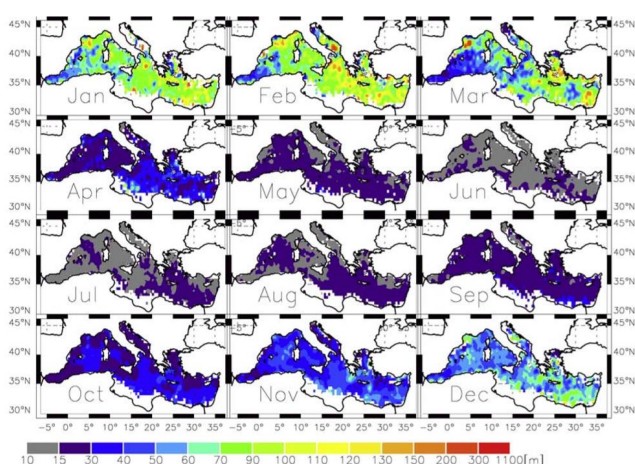





**FIG 2**

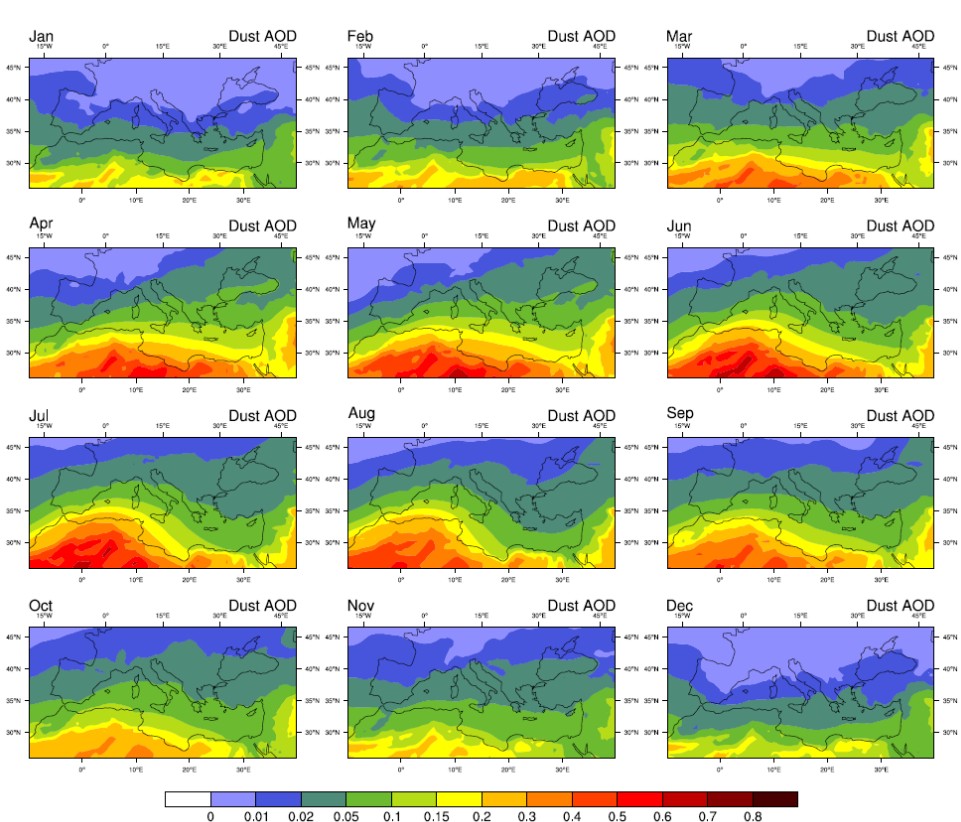





**FIG 3**

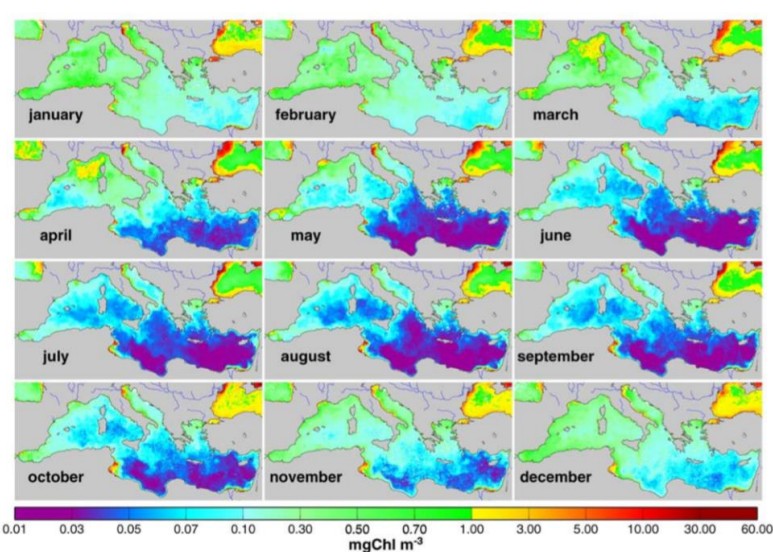


**FIG 4**

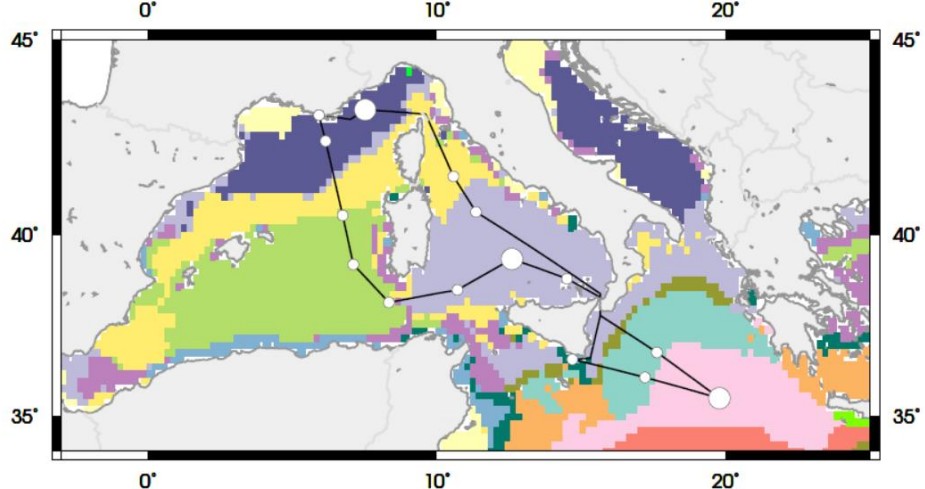







**FIG 5**

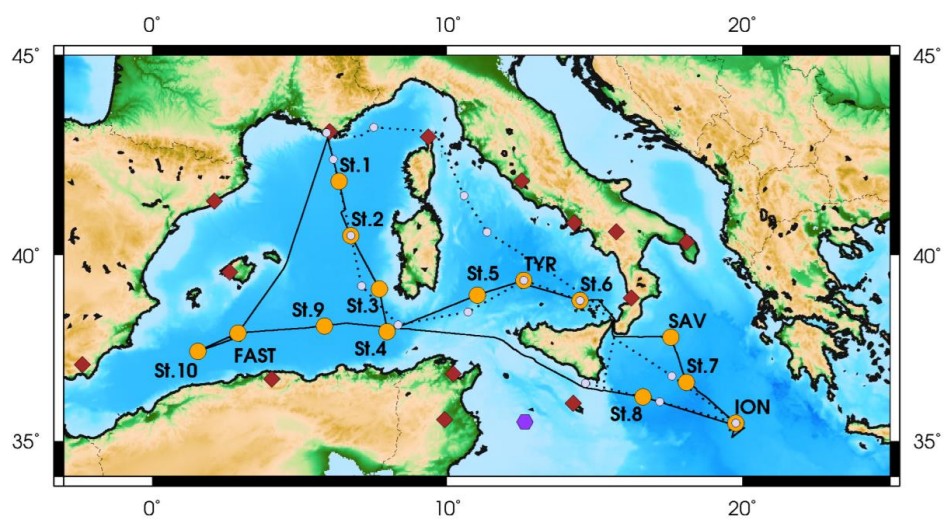


**FIG 6**

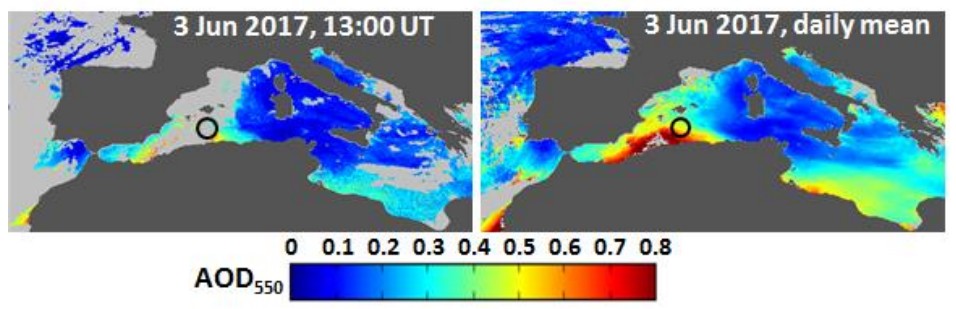




**FIG 7**

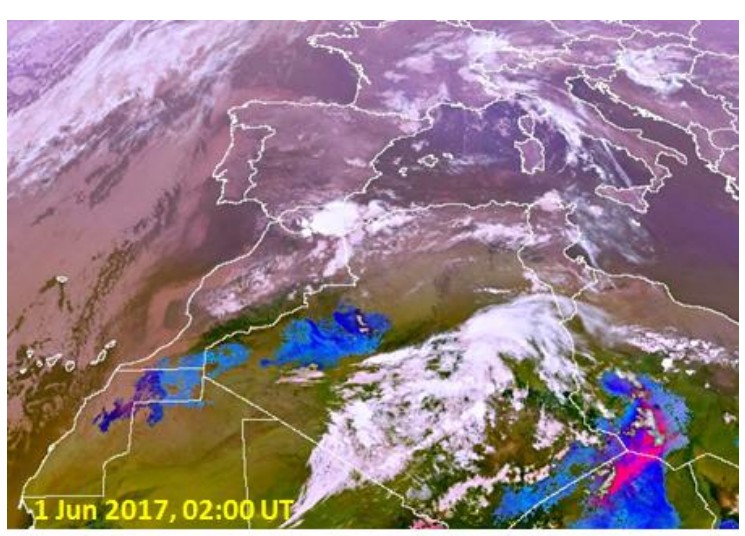


**FIG 8**

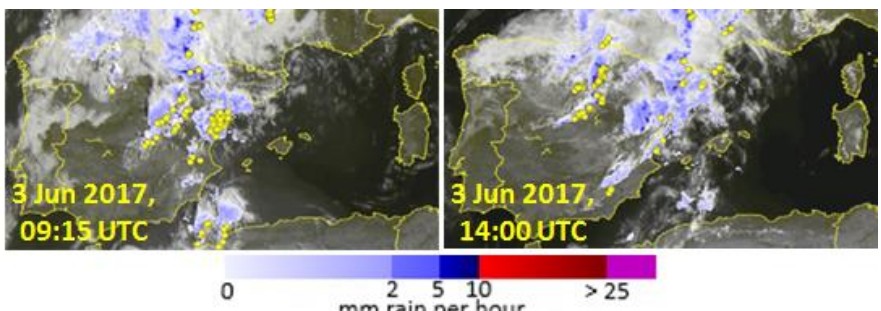







1321                                    **FIG 9**

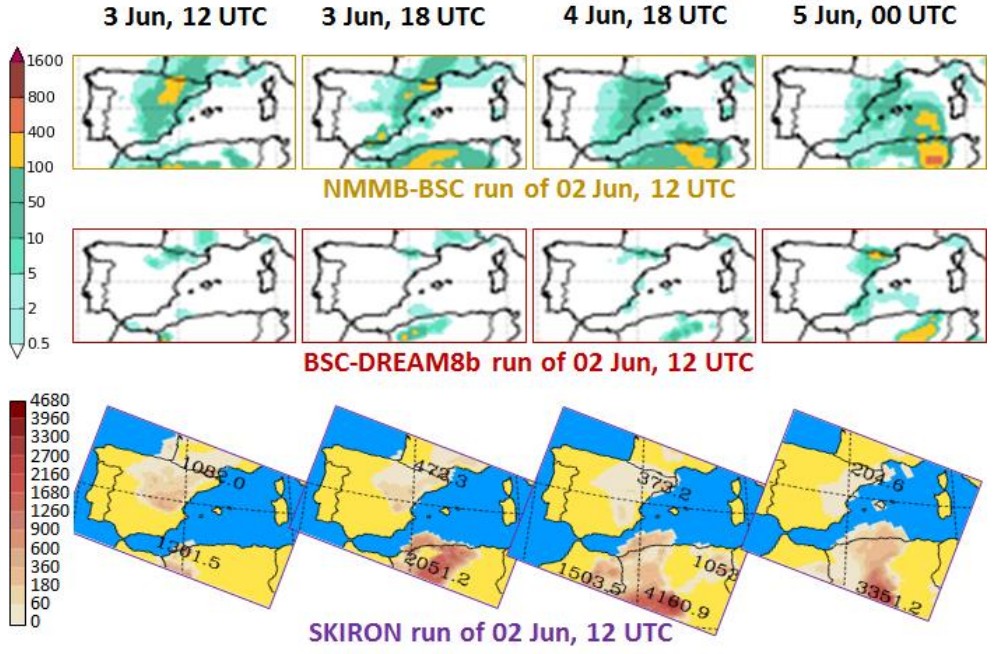


1323                                    **FIG 10**

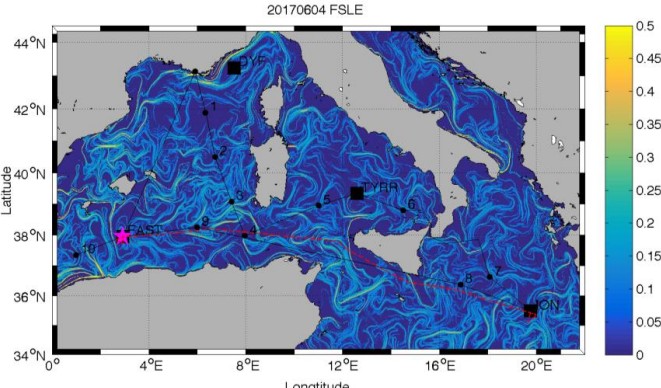





1326                                                 **FIG 11**

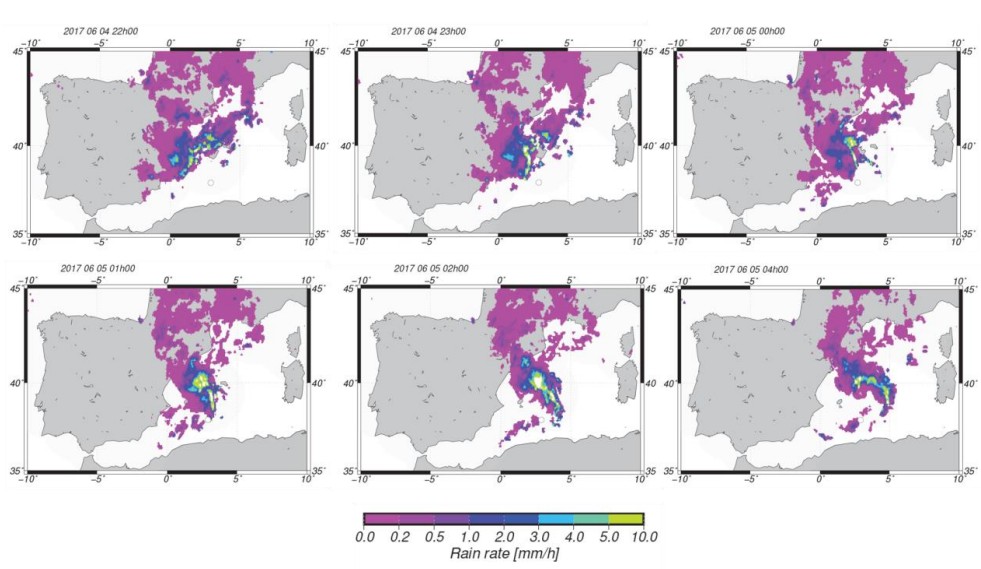






**FIG 12**

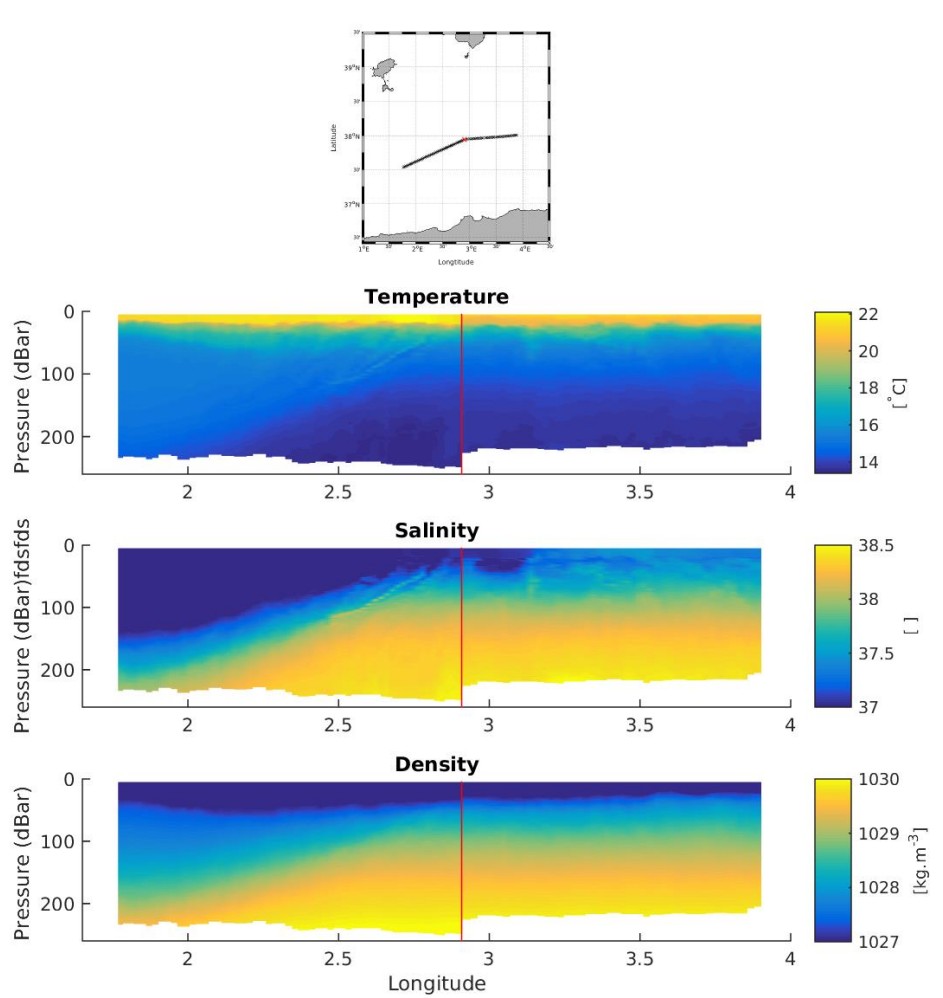





**FIG 13**

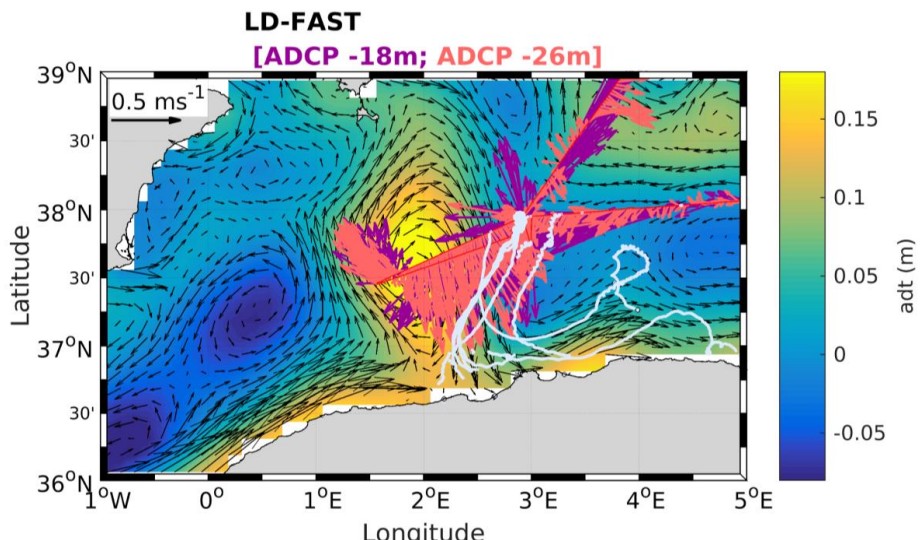


**FIG 14**

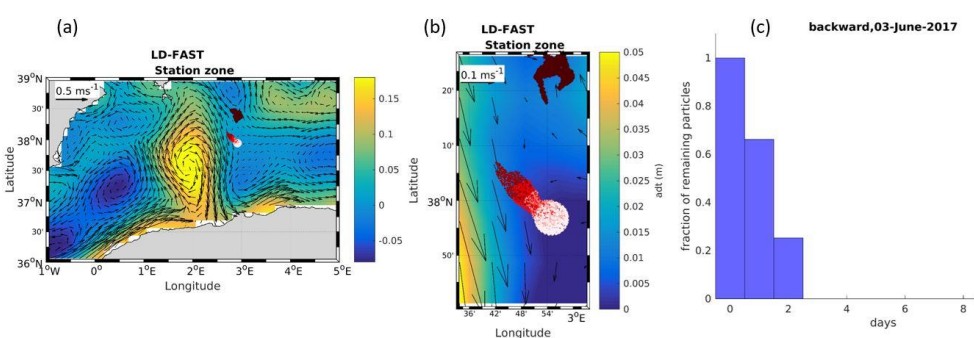






**FIG 15**

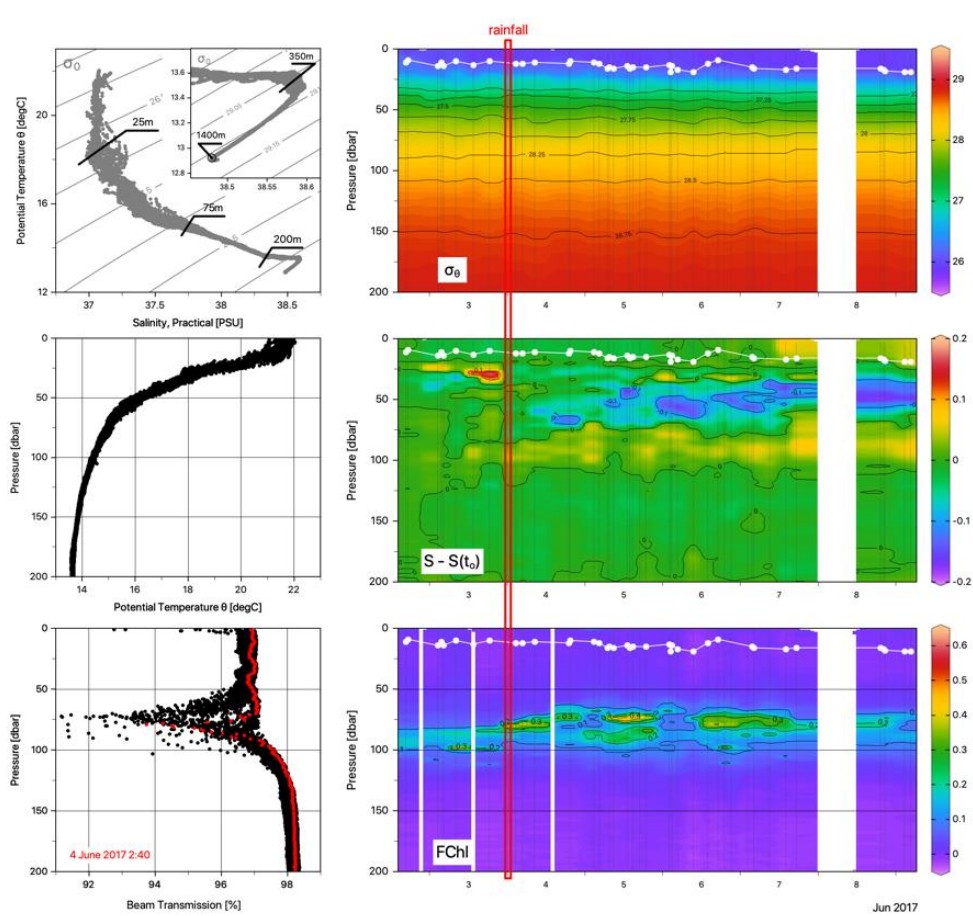


**FIG 16**

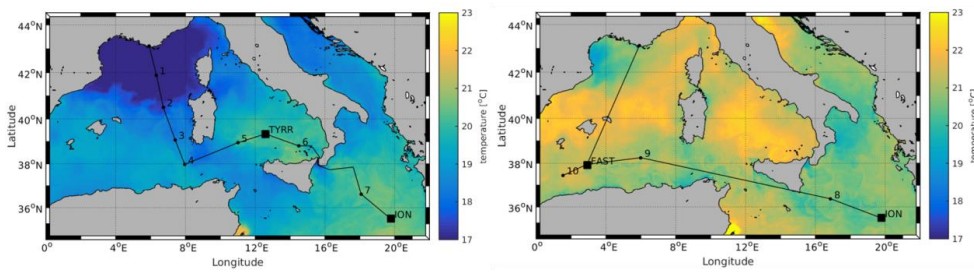







1342                                          **FIG 17**

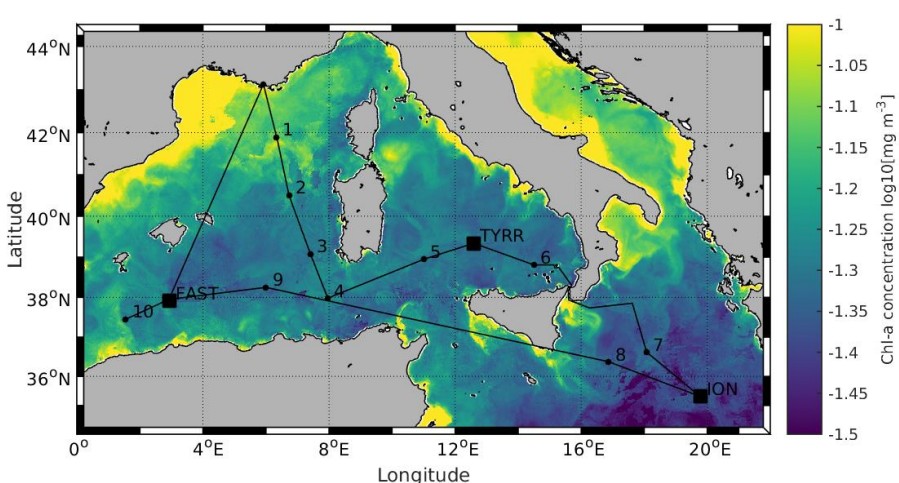





**FIG 18**

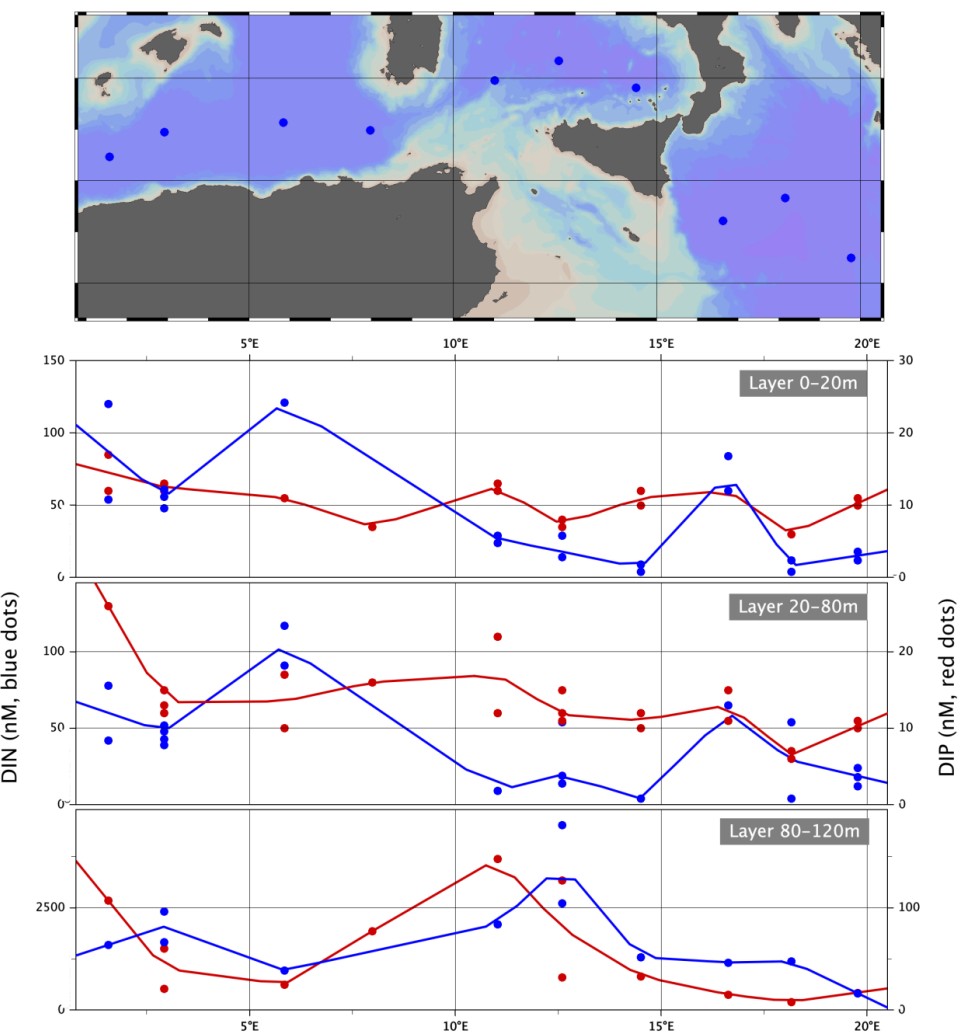
