# Peer review of "Process studies at the air-sea interface after atmospheric deposition in the Mediterranean Sea: objectives and strategy of the PEACETIME oceanographic campaign (May-June 2017)"

_Biogeosciences, 2020_

## Referee Comment (RC1) · Jeffrey W. Krause (Referee) · 13 Apr 2020

As per the journal's main manuscript types, Research Articles (which this manuscript was designated), are to "report substantial and original scientific results within the journal scope."

The manuscript details the impetuous and planning details for the PEACETIME oceanography campaign in the Mediterranean Sea during May-June 2017. My initial impression is this article was to be a sort of "introduction" paper to a special volume of

manuscripts from this program. While I have seen, and been a part of, programs with such introduction papers, the manuscript in its present form introduces the project but does not have any main findings. Specifically, it needs more results, even if it just re-states main points from the collection of papers in the special volume; furthermore, an introduction paper (in my opinion) provides broader context to the importance of these results. Only seven manuscripts in Table 2 are reported as being submitted (actually only four others are submitted because these seven include two publications: Garel et al. 2019; Menna et al. 2019, and this manuscript), while there are over 20 articles listed as "in prep"; it appears too early for a synthesis paper.

Much of the prose is reminiscent of a project proposal which is justifying where/when/why a cruise will be conducted; and unlike a Research Article, I do not see any main conclusions. While it is clear that there was significant planning involved with this cruise to sample just the right conditions, this is a standard practice for PI teams to lead a cruise when it is not sampling some pre-determined spatial grid (CLIVAR, AMT Program, GEOTRACES, etc.). The authors are to be congratulated for successfully sampling the desired conditions, as it clearly took a lot of planning; and I am excited to see these results in the future. However, in my opinion, successfully sampling an event alone does not merit publication as a Research Article.

Below I make some more specific suggestions regarding the science and prose (e.g. numerous grammatical and spelling errors).

Line 22 (also 108), what does "state of the art regarding" refer to? Please correct wording. Line 33, please revise word choice for "valorization" Line 58-59, this is worded like iron is not a trace metal, perhaps revise "iron and other trace metals" Line 68, "metal" not "metals" Line 74, revise wording "also allowing quantifying the export below" Line 75, revise wording "P and N for marine biosphere" Line 76, "several days" requires more context. Several days from the initial part of a multi-day event? After a multi-day event? Line 93-98, this is a long sentence, please break. Line 138, 140, correct to "33-day" and "on-board", respectively Line 149-152, please see general comment.

Line 165, revise wording "dust transport associated to rain period" Line 175, PM10 has not been defined, a casual reader will not be familiar with this term Line 216, "station" should be plural Line 253, correct use of three periods "..." Line 261-266, this is just basic cruise planning Line 285, unlike prior sections, this one is not numbered Line 321, revise to ", on a regular basis," or the sentence could be modified to not need the commas Line 332, correct the units currently "mn" Line 458, correct spelling of "reacher" Line 480, remove extra period between mg and m-2 Line 487, correct the acronym ("MVP" not VMP) Line 528, please consider revising use of "kept lain" Line 547, meaning no biomass was accumulated but what about increase of biological rates (e.g. primary productivity), how were these affected? Line 548, please consider revising wording of "was displayed" Line 550, the increased intensity is an intriguing result. A simple mass balance is merited, e.g. could enough Fe be introduced from the dust lead to this much increase in the deep chlorophyll maximum? Given the spatial separation between surface Fe input and deep chlorophyll increases, I think this would be tenuous; therefore, I think the authors are internal wave idea (line 555-556) is more likely. Line 599-601, this manuscript does not provide any synopsis of these other studies (presumably because a vast majority are still "in prep") and thus reads more like a pre-cruise planning report. Figure 1 – why does the scale jump to 1100 meters? Figure 4 – why is a "proposed" cruise track relevant? As stated above, most cruises not following a pre-determined section (e.g. GEOTRACES) will nearly always have to modify a cruise track.

---

## Referee Comment (RC2) · Anonymous Referee #2 · 15 Apr 2020

The manuscript provides a description of the background, rationale, objectives, planning and execution of the 'PEACETIME' project investigating the influence of atmospheric dust deposition in the Mediterranean Sea. The manuscript provides potentially useful information to the community, both as a component of a special issue presenting the results of this study and hence a reference point for the other manuscripts, and as a case study in how to plan and execute studies of this type, which may be of use for others planning similar research projects in the future. However, I was left wondering whether the manuscript as stands provides the best introduction to the special issue

as there is no attempt at providing synthesis or even a broad overview of the results of the study which will presumably be presented within the other manuscripts. If the authors currently plan to produce an additional overview/introductory manuscript then I would suggest they might consider merging these pieces of work, although it should be noted that the current manuscript is already quite long, in particular currently having 18 individual figures (see below).

If the manuscript is to remain broadly in the current form I wondered whether the authors might broaden the potential interest to a wider community through adding in a section towards the end on 'lessons learnt' or similar. Namely, given the outlined plan and execution, are there recommendations the authors could make on how other groups might undertake and even improve on such a study in the future? Such a section might add value not least because the use of multiple real time observational and forecast data products described does indicate a level of adaptive planning of the cruise track which is perhaps beyond that usually performed. In the absence of some broader discussion of 'lesson learnt', as indicated above, I was personally left wondering whether the material in the manuscript might be considerably condensed and combined with a board overview of the results presented in the other special issue manuscripts.

Irrespective of whether a broader overview manuscript is planned, I note that there were a number of places where additional details might have been provided which were unlikely to compromise other publications. For example, how large was the observed rain deposition event (Line 475)? Related to this section, what does 'lowest most intense' (Line 481) mean?

Some additional general points:

It would be useful to have station locations indicated on more of the figures, in particular figures 7, 8, 9 & 11. It might also be worth compositing these figures as subpanels to enable easier comparison across them. Indeed I found I spent a lot of time moving

backwards and forwards between different figures and the text and I think it would help readability if both the set of figures used and the formatting of these be reconsidered.

There were also a few places where referencing could have been improved. For example, the Okubo-Weiss parameter and Finite Size Lyapunov Exponents are not explained or referenced (Lines 408-409).

Minor corrections: There are a large number of minor typographic and/or grammatical errors throughout the manuscript. Some of these are listed (often as suggested alternative) below, but I will likely have missed many others and the whole manuscript requires a thorough proof read and edit.

Line 22: '. . .we provide a state of the art regarding..' state of the art review?

Line 25: rephrase

Line 32: 'in contrasted areas'

Line 51: 'in addition to these continuous'

Line 67: between less than 10% ?

Line 74 & 74: '. . .also allow quantification of export below. . . for the marine biosphere. . .'

Line 124: 'water column'

Line 148: 'the probability of catching'

Line 165: 'associated with the rainy period'

Line 218: 'lead to changing the planned'

Line 236: 'the relevance of following the initial track was discussed in view of several'

Line 253: sentence appears incomplete

Line 283: maybe 'leading to the decision to start the'

Line 332: 'every 15 minute'?

Line 398: '. . . strategy, with the aim of avoiding regions of. . .'

Line 409: 'exponent'

Line 420: 'the presence of a dust plume'

Line 436: 'On the 31st of May. . . between the islands of Sicily and Sardinia'

Line 446: 'On the 1st of June'

Line 454: during the first half of June 5th

Line 458: 'richer in'

Line 489: MVP is not defined here on first use

Line 526: '. . .due to the rainfall. . .'

Line 528: needs rephrasing

Line 546: intermittent signals

Line 592: '. . .productivity and high probability of wet dust deposition.'

Line 1246: 'allowing detection of'

---

## Author Comment (AC1) · 9 Jun 2020

We thanks you for your constructive suggestions and comments.

Both reviewers found the paper too descriptive and requested that more results should be presented in order to show how the project allowed some progress on the scientific questions. A dedicated section is now available highlighting the main results. Also, since the initial submission of our paper, the following progress was made regarding the papers from PEACETIME results:

- 2 papers have been accepted to the Special Issue (Tovar-Sanchez et al. and Taillandier et al.)

- 4 papers have been submitted to the Special Issue (Freney et al., Trueblood et al., Feliu et al., Gazeau et al.)

- 4 papers are published in other journals (Bressac et al., Nat. Geosc. And Whitby et al., in GRL, Garel et al., 2019, Menna et al., 2019)

- 1 paper has been submitted to PNAS (Sellegri et al.)

- 14 other papers are still in preparation for this special issue with submissions in June and July.

- We decided also to remove from this list the papers that are in prep for a different journal

We choose in our first version to provide a full description of the decision tools and we agree that leaving that section and adding a long new section summarizing the content of the papers that are/will be presented in the SI would make a too long paper. That section (along with 5 figures) is now presented as Supplemented Material. Because we removed part of the text, the outline changed a bit also. A marked-up manuscript version is also available.

Specific suggestions :

Line 22 (also 108), what does "state of the art regarding" refer to? Please correct wording.
Sentence changed

Line 33, please revise word choice for "valorization"
Sentence removed. The abstract was changed accordingly

Line 58-59, this is worded like iron is not a trace metal, perhaps revise "iron and other trace metals"
Changed

Line 68, "metal" not "metals"
Changed

Line 74, revise wording "also allowing quantifying the export below"
Changed « equipped with sediment traps »

Line 75, revise wording "P and N for marine biosphere"

« for marine biosphere » was removed

Line 76, "several days" requires more context. Several days from the initial part of a multi-day event? After a multi-day event?
« after the rain event was simulated » was added

Line 93-98, this is a long sentence, please break.
Done

Line 138, 140, correct to "33-day" and "on-board", respectively
Done

Line 149-152, please see general comment.
The end of the introduction was changed. Please check the marked-up manuscript version.

Line 165, revise wording "dust transport associated to rain period"
« period » removed

Line 175, PM10 has not been defined, a casual reader will not be familiar with this term
« particles with diameter smaller than 10 µm (PM10) » have been added

Line 216 "station" should be plural
Done

Line 253, correct use of three periods ": : :"
Done

Line 261-266, this is just basic cruise planning
We do not really agree on that as the R/V had to move 800 km (450 nm) far from its initial route and be at its new position on time to sample a rain event, which is not very common in oceanographic cruises. Moreover this had to be done without delay in order to be at station before the event impacted the targeted area. Anyway, a long part of all that section is now moved to Supplementary Material (8 pages), we hope that the reading will get easier that way.

Line 285, unlike prior sections, this one is not numbered
The section order and numbering is now changed to have a stronger focus – as requested by both reviewers – on outputs.

Line 321, revise to ", on a regular basis," or the sentence could be modified to not need the commas
Done
Note that all that section is now in the Supplementary Material

Line 332, correct the units currently "mn"
Done
Note that all that section is now in the Supplementary Material

Line 458, correct spelling of "reacher"

The whole sentence was changed for « where the influence of Atlantic waters characterized by different nutrients pattern than Mediterranean waters should be limited compared to the more western Alboran Sea ».

Line 480, remove extra period between mg and m-2
Done

Line 487, correct the acronym ("MVP" not VMP)
Done

Line 528, please consider revising use of "kept lain"
Sentence changed : "The density horizons being maintained along isobars in the upper layer, sign the absence of geostrophic perturbations during the long station"

Line 547, meaning no biomass was accumulated but what about increase of biological rates (e.g. primary productivity), how were these affected?

At FAST, after the rain, we observed first an increase in nutrients (DIN and DIP) in the mixed layer followed by a decrease in the 24h (DIN) and 48h (DIP). This was concomitant to increases in PP and BP. Yet, no increase in biomass (from pigments) was observed. But we know from previous experiments that stocks are not good proxies to evaluate the impact (for ex. grazing can hide the increase of biomass and visible changes in zooplankton community followed the dust event (Feliu et al., in revision). The N and P demand to fulfil this increase in PP and BP was calculated and compared to the decrease in nutrients and we concluded that the atmospheric deposition could explain these metabolic fluxes changes (van Wambeke et al., in prep). Importantly, we also checked the fluxes from below and found that diapycnal flux of phosphate to the mixed layer was particularly weak at FAST and was 2 order of magnitude lower than atmospheric soluble flux for P for example (in Pulido-Villena et al., in prep). Vertical diffusion fluxes from the interior into the depleted layer (across nutriclines) were much higher (Taillandier et al., 2020; Pulido-Villena et al. in prep); however, those nutrients were not injected up to the shallower mixed layer that was rather directly impacted by atmospheric deposition. (see section 6)

Line 548, please consider revising wording of "was displayed"
 « A deep chlorophyll maximum of about 20-m thickness was located at the base of the thermocline (about 75 m). »

Line 550, the increased intensity is an intriguing result. A simple mass balance is merited, e.g. could enough Fe be introduced from the dust lead to this much increase in the deep chlorophyll maximum? Given the spatial separation between surface Fe input and deep chlorophyll increases, I think this would be tenuous; therefore, I think the authors are internal wave idea (line 555-556) is more likely.
As described in Bressac et al. in prep, the DFe concentrations could be well followed after the dust event. However, DFe was not limiting and high concentrations are usually found in the Med sea due to the accumulation from dust deposition during the stratification period as shown for PEACETIME and from previous studies. Atmospheric input does not impact DCM, too deep and well below the stratification. (see also our comment on diffusion at different depths), so our hypothesis about internal wave seems correct.

Line 599-601, this manuscript does not provide any synopsis of these other studies (presumably because a vast majority are still "in prep") and thus reads more like a pre-cruise planning report.

The whole section 6 is now dedicated to an overview of the results and associated papers.

Figure 1 – why does the scale jump to 1100 meters?

The maximum values of the MLD are observed in February and in February–March for the Gulf of Lions and the Southern Adriatic Sea respectively, which are regions of Deep Water Formation.

Figure 4 – why is a "proposed" cruise track relevant? As stated above, most cruises not following a pre-determined section (e.g. GEOTRACES) will nearly always have to modify a cruise track.

Indeed, cancel/add or shift a station is common but we believe that, a 800 km 'deviation' is not that common. Rerouting a ship from such a distance based on dust plume/rain forecast in order to catch an atmospheric deposition event was something innovative, wet deposition sampling on board R/Vs are often opportunist.

---

## Author Comment (AC2) · 9 Jun 2020

We thanks you for your constructive suggestions and comments.

Both reviewers found the paper too descriptive and requested that more results should be presented in order to show how the project allowed some progress on the scientific questions. A dedicated section is now available highlighting the main results. Also, since the initial submission of our paper, the following progress was made regarding the papers from PEACETIME results:

- 2 papers have been accepted to the Special Issue (Tovar-Sanchez et al. and Taillandier et al.)

- 4 papers have been submitted to the Special Issue (Freney et al., Trueblood et al., Feliu et al., Gazeau et al.)

- 4 papers are published in other journals (Bressac et al., Nat. Geosc. and Whitby et al., in GRL, Garel et al., 2019, Menna et al., 2019)

- 1 paper has been submitted to PNAS (Sellegri et al.)

- 14 other papers are still in preparation for this special issue with submissions in June and July.

- We decided also to remove from this list the papers that are in prep for a different journal

We choose in our first version to provide a full description of the decision tools and we agree that leaving that section and adding a long new section summarizing the content of the papers that are/will be presented in the SI would make a too long paper. That section (along with 5 figures) is now presented as Supplementary Material. Because we removed part of the text, the outline changed a bit also. A marked-up manuscript version is also available.

Irrespective of whether a broader overview manuscript is planned, I note that there were a number of places where additional details might have been provided which were unlikely to compromise other publications. For example, how large was the observed rain deposition event (Line 475)?
Please note that the rain rates are depicted on Figure 5. The following sentence was added: "This rain was part of a massive rain front covering ~80 000 km$^2$.This front extended from the coast of Spain to the south of the FAST station area, with rain rate reaching 10 mm h-1 (Figure 5)"

Related to this section, what does 'lowest most intense' (Line 481) mean?
According to the classification of Vincent et al. (2016), the most intense dust deposition (MIDD) event is defined by considering all the samples for which the weekly deposition flux is greater than the geometric mean + the geometric standard deviation for a given station. Here, we considered the "Majorca Island" station of Vincent et al. as a reference station because it is the closest from the FAST station: comparing with that reference, our sampled wet dust deposition was in the low range of MIDD reported in the region.
So we replaced by "in the low range of most intense dust deposition event (MIDD)"

Some additional general points: It would be useful to have station locations indicated on more of the figures, in particular figures 7, 8, 9 & 11.

Those figures are now in SI. The position of the FAST station is now reported on each.

It might also be worth compositing these figures as subpanels to enable easier comparison across them. Indeed I found I spent a lot of time moving backwards and forwards between different figures and the text and I think it would help readability if both the set of figures used and the formatting of these be reconsidered. There were also a few places where referencing could have been improved. For example, the Okubo-Weiss parameter and Finite Size Lyapunov Exponents are not explained or referenced (Lines 408-409).
We believe that the reading will be easier now that the whole section and the associated figures are moved to SI.

Minor corrections: There are a large number of minor typographic and/or grammatical errors throughout the manuscript. Some of these are listed (often as suggested alternative) below, but I will likely have missed many others and the whole manuscript requires a thorough proof read and edit.
Thank you for your editing. We did a careful proofread and we hope this aspect was improved.

Line 22: ': : :we provide a state of the art regarding..' state of the art review?
Same as Rev1, this sentence was changed

Line 25: rephrase
The whole abstract was modified. Those sentences have been changed
Line 32: 'in contrasted areas'
The whole abstract was modified. Those sentences have been changed

Line 51: 'in addition to these continuous'
Changed

Line 67: between less than 10% ?
This was rephrased,

Line 74 & 74: ': : :also allow quantification of export below: : : for the marine biosphere: : :'
The whole sentence was rephrased

Line 124: 'water column'
Done

Line 148: 'the probability of catching'
Done

Line 165: 'associated with the rainy period'
done

Line 218: 'lead to changing the planned'
Done

Line 236: 'the relevance of following the initial track was discussed in view of several'
This sentence was removed

Line 253: sentence appears incomplete
Corrected

Line 283: maybe 'leading to the decision to start the'
Done

Line 332: 'every 15 minute'?
Done (this section is now in SI)

Line 398: ': : : strategy, with the aim of avoiding regions of: : :'
Done

Line 409: 'exponent'
Done

Line 420: 'the presence of a dust plume'
A part of that section was shortened

Line 436: 'On the 31st of May: : : between the islands of Sicily and Sardinia'
Done

Line 446: 'On the 1st of June'
Done

Line 454: during the first half of June 5th
Done

Line 458: 'richer in'
Sentence was rephrased

Line 489: MVP is not defined here on first use
Done

Line 526: ': : :due to the rainfall: : :'
Done

Line 528: needs rephrasing
Done

Line 546: intermittent signals
Done

Line 592: ': : :productivity and high probability of wet dust deposition.'
Done

Line 1246: 'allowing detection of'
Done

---

## Editor Comment (EC1) · Jan-Berend Stuut (Editor) · 15 Jun 2020

The authors have submitted their response to the two reviewers' assessments of the original manuscript and submitted a revised manuscript with changes according to the reviewers' questions and suggestions. Unfortunately, the revised manuscript could not be found on the Copernicus server. Therefore, I have requested this revised manuscript from the authors and have noted that they have taken the reviewers' suggestions to heart and have changed the manuscript considerably. Where initially the manuscript was a not-so-organised summing up of initial or anticipated results

of the PEACETIME cruise, in the revised manuscript the authors have done a great job in taking the reader on board the ship (R/V Pourquois Pas?) during the cruise and describing what happened during this very exciting expedition. In that sense, this manuscript is an excellent introduction for all the following manuscripts that describe parts of the results from the cruise. The authors have now clearly sketched what a tremendous enterprise the PEACETIME cruise has been, involving so many scientific disciplines both on board the ship as well as synchronously on land measuring, modelling and forecasting dust outbreaks. Indeed, the expedition managed to measure a score of different settings, allowing the detailed study of various types of deposition (e.g., dry vs wet) and different types of oceanographic settings. The authors describe the cruise and all the ongoing different types of experiments in detail, which forms a great introduction to the special volume as well as to the forthcoming manuscripts, which are all cited as either 'in preparation' or 'submitted'. My recommendation of the revised manuscript is "accept after minor revisions" (some small changes marked in the annotated revised manuscript), as from my assessment of the authors' response to the two reviewers and the revised version of the manuscript I can only conclude that the authors have significantly improved the manuscript. However, I propose to postpone this acceptance until either this special volume is formally closed or until all manuscripts have gone through the review process. That way, the present citations to manuscripts that are yet to be submitted can be completed and/or changed in case these manuscripts were not accepted after all.

Please also note the supplement to this comment:
https://www.biogeosciences-discuss.net/bg-2020-44/bg-2020-44-EC1-supplement.pdf

**Supplement:**

**Process studies at the air-sea interface after atmospheric deposition in the Mediterranean Sea: objectives and strategy of the PEACETIME oceanographic campaign (May-June 2017)**

Cécile Guieu1, Fabrizio D'Ortenzio1, François Dulac2, Vincent Taillandier1, Andrea Doglioli3, Anne Petrenko3, Stéphanie Barrillon3, Marc Mallet4, Pierre Nabat4, Karine Desboeufs5

[revised manuscript text omitted]

**6.2 Biogeochemical and physical features from instruments**

Below the sea surface, a number of observations acquired with instruments deployed during the cruise (see Table 2) or thanks to autonomous floats launched well before and during the campaign, allowed to depict interesting features regarding transport of particles by water masses, nutrients dynamics and biological production. A high resolution sampling of particulate matter distribution revealed sharp horizontal gradients in the central Ionian Sea, particulate matter originating from the Western basin (Modified Atlantic water mass) and Adriatic Sea (Ionian Sea Water) **
[revised manuscript text omitted]

---

## Author Comment (AC3) · 29 Jun 2020

We would like to thank the editor for his positive response to the consideration of our revised manuscript. We would also like to thank him for his careful reading and the requested editorial corrections have been taken into account. The editor proposed "to postpone the acceptance of the article until either the special volume is formally closed or until all manuscripts have gone through the review process". This could lead to a fairly substantial delay in the acceptance of our article, as recently, the authors of 4 manuscripts (among the 23 expected from PEACETIME for the Special Issue) informed

us of a delay in the submission of their article (linked to the covid crisis, especially for the teacher-researchers who were not able to devote themselves as planned to their research but had to devote all their time to distance learning). These submissions are now scheduled for September, which postpones the acceptance of our article by at least 2-3 months, which seems a bit long to us. Off course, we leave that decision to the journal.

—————————————————————

---

## Editor Decision (ED2)

[revised manuscript text omitted]

|      | Table 2 Contract                         |                                                                 |           |          |          |                     |                          |                      |            |             |                                                                                                      |
|------|------------------------------------------|-----------------------------------------------------------------|-----------|----------|----------|---------------------|--------------------------|----------------------|------------|-------------|------------------------------------------------------------------------------------------------------|
|      | 1 able 2 (contr                          | nuea)                                                           |           |          | -        |                     |                          | -                    |            |             |                                                                                                      |
|      | 3 times                                  | Drifting mooring                                                | [11]      |          |          |                     |                          | х                    | X          | Х           | Bressac et al., (2019); Bressac et al., in prep.                                                     |
|      | 3 times                                  | Sediment Cores                                                  | [12]      |          |          |                     | _                        | x                    | x          | x           | Brandt et al., submitted; Brandt et al., in prep.                                                    |
|      | 20 drifters                              | SVP (Surface Velocity
Program) drifters                      | [13]      |          |          |                     |                          | ×                    | x          |             | Menna et al., 2019. Guieu et al., this paper; Berline et al.in prep., Desboeufs et al., in prep. (b) |
|      | a total of 1000
profiles (0-
300m) | ) Moving Vessel Profiler
(begining and end long
stations) | [14]      | betwee   | n the sh | ort statio
frequ | ins and in
tent as po | n the lon
ossible | g stations | s area as   | Guieu et al., this paper; Berline et al. in prep.                                                    |
|      | 2
deployments,
1 recovery          | Biogeochemical ARGO float                                       | [15]      |          |          |                     | ×                        |                      | ×          |             | Barbieux et al., in prep. ; Taillandier et al., submitted                                            |
| 1156 |                                          |                                                                 |           |          |          |                     |                          |                      |            |             |                                                                                                      |
| 1157 | 1] Atmosphe                              | ric sampling was carried ou                                     | t throu   | ghout t  | he tran  | isect us            | ing PE(                  | GASU                 | S dedica   | ated mo     | bile platform (PortablE Gas and Aerosol Sampling                                                     |
| 1158 | UnitS, Form                              | enti et al., 2019) to monito                                    | or conti  | snonu    | air ga   | seous c             | sodmo                    | ition (]             | VOX, S(    | $O_2, O_3,$ | CO 2 , CO, VOC), physico-chemical properties of                                           |
| 1159 | aerosol parti                            | cles (mass and number con                                       | ıcentra   | tion, £  | size-di  | stributi            | on, che                  | mical                | compos     | ition a     | nd nutrients contents), parameters of atmospheric                                                    |
| 1160 | dynamics su                              | ch as the boundary layer, an                                    | nd radi   | utive pɛ | tramet   | ers (inc            | ident ra                 | adiatio              | n, optica  | al thick    | ness, optical properties of the particles).                                                          |
| 1161 | [2] Rain sam                             | pling of two events that occ                                    | urred c   | luring t | he cru   | iise. Th            | e on-lin                 | ne filtra            | tion col   | lector v    | was used to determine the dissolved and particulate                                                  |
| 1162 | composition                              | of rain, including major and                                    | l trace 1 | netals ( | (Al, Βε  | ı, Cd, C            | 0, Cr, (                 | Cu, Fe,              | Mo, Mn     | ı, Ni, Pł   | o, Sr, Ti, V, Zn), atmospheric inorganic compounds                                                   |
| 1163 | (sulfate, chlc                           | rrure, Na, Mg, K, Ca, ) and                                     | l dissol  | lved nı  | utrient  | s (phos]            | phate, 1                 | nitrate,             | ammon      | ium).       |                                                                                                      |
| 1164 | [3] An innov                             | ative system of continuous '                                    | "clean'   | dund     | ing act  | tivated l           | by a lar                 | ge peri              | staltic p  | oump cc     | nnected to a tube plunged at 5 m under the surface                                                   |
| 1165 | seawater ins.                            | ide a TraOcean was set up.                                      | . The     | vater v  | vas co   | nveyed              | in a d                   | edicate              | d labor    | atory a     | nd distributed to several instruments to assess its                                                  |

[revised manuscript text omitted]